# Dual interfacial engineering of a Chevrel phase electrode material for stable hydrogen evolution at 2500 mA cm$^{-2}$

Heming Liu[1,2,6], Ruikuan Xie[3,6], Yuting Luo[1,2], Zhicheng Cui[2], Qiangmin Yu[1,2], Zhiqiang Gao[4,5], Zhiyuan Zhang[1,2], Fengning Yang[1,2], Xin Kang[1,2], Shiyu Ge[1,2], Shaohai Li [1,2], Xuefeng Gao [4,5], Guoliang Chai[3], Le Liu [2] & Bilu Liu [1,2] ✉

Constructing stable electrodes which function over long timescales at large current density is essential for the industrial realization and implementation of water electrolysis. However, rapid gas bubble detachment at large current density usually results in peeling-off of electrocatalysts and performance degradation, especially for long term operations. Here we construct a mechanically-stable, all-metal, and highly active $CuMo_6S_8$/Cu electrode by in-situ reaction between $MoS_2$ and Cu. The Chevrel phase electrode exhibits strong binding at the electrocatalyst-support interface with weak adhesion at electrocatalyst-bubble interface, in addition to fast hydrogen evolution and charge transfer kinetics. These features facilitate the achievement of large current density of 2500 mA cm$^{-2}$ at a small overpotential of 334 mV which operate stably at 2500 mA cm$^{-2}$ for over 100 h. In-situ total internal reflection imaging at micrometer level and mechanical tests disclose the relationships of two interfacial forces and performance of electrocatalysts. This dual interfacial engineering strategy can be extended to construct stable and high-performance electrodes for other gas-involving reactions.

The extensive use of fossil fuel has caused environmental pollution and energy crisis. Especially, carbon emissions from fossil fuel contribute ~65% of the total global emissions of greenhouse gases[1] and it is thus urgent to develop green energy. Due to the superiorities of high energy efficiency, high mass-energy density, and zero-carbon emissions[2], green hydrogen produced by electrochemical water splitting is promising as future clean energy carrier. However, electrocatalysts applied in industrial water splitting have to suffer from harsh conditions, such as large current density, long working period, high pressure and elevated temperature[3–6], which bring challenges to the mechanical stability of electrodes and making widespread implementation of

electrolysis difficult. The mechanical stability of an electrode is mostly determined by two interfaces, i.e., the electrocatalyst-support interface and electrocatalyst-bubble interface. The detachment of many gas bubbles at large current density would produce a strong electrocatalyst-bubble adhesion force[7–10], which can peel off the electrocatalysts when it is larger than the binding force between the electrocatalyst and support. Therefore, it is important to enhance mechanical stability of electrodes by enhancing electrocatalyst-support and weakening electrocatalyst-bubble interfacial forces.

Along this direction, the most common way to enhance the adhesion of electrocatalysts on supports is using binders such as

[1]Shenzhen Geim Graphene Center, Tsinghua-Berkeley Shenzhen Institute & Shenzhen International Graduate School, Tsinghua University, Shenzhen 518055, P. R. China. [2]Institute of Materials Research, Shenzhen International Graduate School, Tsinghua University, Shenzhen 518055, P. R. China. [3]State Key Laboratory of Structural Chemistry, Fujian Institute of Research on the Structure of Matter, Chinese Academy of Sciences, Fuzhou 350002, P. R. China. [4]Functional Materials and Interfaces Lab, Suzhou Institute of Nano-Tech and Nano-Bionics, Chinese Academy of Sciences, Suzhou 215123, P. R. China. [5]School of Nano-Tech and Nano-Bionics, University of Science and Technology of China, Hefei 230026, P. R. China. [6]These authors contributed equally: Heming Liu, Ruikuan Xie. ✉e-mail: bilu.liu@sz.tsinghua.edu.cn

Nafion, but it usually cannot provide an adhesion force that is large enough to stand bubble bombardment[11]. Moreover, the use of binder may also block active sites, reduce ionic conductivity and make the electrode aerophilic[12–15], which are harmful to reaction kinetics and bubble detachment. To avoid these side effects, many methods to in situ grow electrocatalysts on supports have been developed to construct binder-free self-supporting electrodes[16–19]. Such in situ growth can well control the morphology of electrode to promote bubble detachment and reduce the electrocatalyst-bubble adhesion force[20–22]. Wet chemical synthesis and vapor phase deposition are two common in situ growth methods. So far, most of the thus grown electrocatalysts adhere to the supports mainly via weak physical or chemical interactions, including electrostatic adsorption, mechanical interlocking of porous structure and intermolecular attraction induced by van der Waals force[23,24]. Such weak interface between electrocatalyst and support is usually difficult to stand bubbling[25,26], and leads to poor stability of electrode. In addition, in the case of widely-used semiconducting electrocatalysts[27,28], the contact between semiconducting electrocatalyst and metal support commonly shows Schottky barrier at interface, bringing high contact resistance and retarding reaction kinetics[29–32]. Therefore, developing synthesis method that can produce metallic, highly active electrode with strong electrocatalyst-support interface and weak electrocatalyst-gas interface is important for gas-involving electrocatalysis.

Here, we develop a mechanically stable all-metal $CuMo_6S_8$/Cu electrode, of which Chevrel phase $CuMo_6S_8$ derived from 2H-phase $MoS_2$ and in-situ grown on Cu foam. In previous work, nano-Chevrel phase[33] and layered $Cu_2S$/$Cu_{2.76}Mo_6S_8$/$MoS_2$ heterojunction[34] were synthesized to study their HER activity. These catalysts delivered geometrical current densities of 20 and 100 mA cm$^{-2}$, which are relatively small. By optimizing the electrode structure, high performance and good stability of Chevrel phase catalysts at large current density could be realized. In this work, our $CuMo_6S_8$/Cu electrode has a strong electrocatalyst-support interfacial binding force, a weak electrocatalyst-bubble adhesion force, and all-metal property, which effectively avoid peeling-off of electrocatalysts and improve reaction kinetics at large current density. As a result, the electrode has superior HER performance with a small overpotential of 334 mV to reach 2500 mA cm$^{-2}$ and operates stably at 2500 mA cm$^{-2}$ for over 100 h. In addition, we use the in-situ total internal reflection imaging method to visualize the peeling-off of electrocatalysts at large current density with a micrometer spatial resolution, showing that the peeling-off degree of electrocatalysts on $CuMo_6S_8$/Cu electrode is four times smaller than that on Pt/C electrode. Such distinguished stability of $CuMo_6S_8$/Cu electrode is because it shows an interfacial binding force twice of Pt/C electrode and an electrocatalyst-bubble adhesion force half of Pt/C electrode. The highly active sites and metallic properties of $CuMo_6S_8$ also guarantee superior HER performance at large current density. The dual interfacial engineering developed here can in principle be extended to construct stable and high-performance electrodes for other gas-involving reactions.

## Results

### Preparation and characterization of the Chevrel phase $CuMo_6S_8$/Cu electrode

The $CuMo_6S_8$/Cu electrode is prepared by intermediate-assisted grinding exfoliation of bulk $MoS_2$ to produce 2D $MoS_2$[35], followed by loading them on Cu foam and high-temperature annealing (Fig. 1a, Methods and Supplementary Fig. 1). $MoS_2$ nanoflakes with an average thickness of 36 nm are obtained (Supplementary Figs. 2–3), and X-ray diffraction (XRD) and Raman spectroscopy results show that its composition and phase show negligible change after exfoliation (Supplementary Fig. 3). Before annealing, there are only physical contact and weak electrostatic adsorption between the $MoS_2$ nanoflakes and Cu foam support. The Schottky barrier will form at the interface because

of metal-semiconductor contact, which will impede electron transfer from support to electrocatalyst. To strengthen the interfacial binding and eliminate this Schottky barrier, the electrode is annealed at 750 °C under Ar and $H_2$ atmosphere. This process may also lead to the Wenzel-state wetting property and realize fast gas bubble detachment in terms of macro-performance (Fig. 1b). After annealing, scanning electron microscopy (SEM) images show that the morphology of materials changes from agglomerated nanoflakes to porous structure, which composed of nanoflakes or nanoparticles and a few nanowires. (Supplementary Figs. 2d and 4). The morphological diversity of the material may be due to the low melting point and high-temperature diffusion of Cu. Energy dispersive X-ray spectroscopy elemental mappings of a cross-section of the electrode indicate that Cu atoms react with $MoS_2$ (Fig. 1c–f) and in-situ form Chevrel phase $CuMo_6S_8$ during annealing, which is also confirmed by high-resolution transmission electron microscopy (HRTEM) image. Figure 1g shows the cross-sectional HRTEM image of the $CuMo_6S_8$/Cu electrode prepared by focused ion beam cutting. The zoom-in view of Fig. 1h shows the interface between $CuMo_6S_8$ layer and Cu substrate. The lattice spacings of 0.22 nm and 0.21 nm correspond to (131) and (111) planes of $CuMo_6S_8$ and Cu, respectively. The insets are the corresponding FFT patterns, which also show the crystalline nature of $CuMo_6S_8$ and Cu. The dotted line indicates the interface between them, which is smooth, indicating a close contact between $CuMo_6S_8$ and Cu with robust interface. Both SEM and TEM characterization demonstrate that there is a well-defined robust interface between $CuMo_6S_8$ and Cu. In addition, HRTEM images of other crystal facets of $CuMo_6S_8$ are provided in Supplementary Fig. 5. The XRD patterns show that after annealing, the 2H-phase $MoS_2$ is completely converted into Chevrel phase $CuMo_6S_8$ (PDF #34-1379) with the main diffraction peaks of (101) at 13.7°, (003) at 25.6°, and (131) at 40.8° (Fig. 1i and Supplementary Fig. 7, 8). The peaks from Cu and $Cu_2O$ are also observed because electrocatalysts are loaded on Cu foam. The zoom-in inspection of the peaks in 2 theta of 10–20° is shown in Supplementary Fig. 6. Besides, Raman peaks including $E_g$ (126, 145, 360, 384 cm$^{-1}$) and $A_g$ (202, 220, 285 cm$^{-1}$) are found, which correspond to the characteristic peaks of Chevrel phase (Supplementary Fig. 9)[36]. The stoichiometric ratio of Mo to S is 0.73, as measured by the inductively coupled plasma optical emission spectrometer (ICP-OES, Supplementary Table 1), suggesting the formation of pure $CuMo_6S_8$ with a theoretical Mo to S ratio of 0.74. In addition, X-ray photoelectron spectroscopy (XPS) survey spectrum is also implemented to detect the chemical bonding of the $CuMo_6S_8$ layer that cover the surface of Cu foam, and shows that there are five elements (Cu, Mo, S, C and O) in the sample (Supplementary Fig. 10). The Mo $3d$ spectrum shows that two peaks are located at 231.1 eV and 227.9 eV from $3d_{3/2}$ and $3d_{5/2}$, which are originated from Mo-S bonds in $CuMo_6S_8$[37] (Fig. 1j). The deconvoluted peaks of S $2p$ spectrum can be assigned to S-Mo and S-Cu bonds, which locate at 162.9 eV and 161.6 eV for S-Mo bonds, 164.6 eV and 162.4 eV for S-Cu bonds[38,39] (Fig. 1k). Taken together, the above results show that the Chevrel phase $CuMo_6S_8$ is synthesized by in-situ reaction of $MoS_2$ and Cu foam, which binds tightly to the support by chemical bonding.

### HER performance at large current density

Next, we study the HER performance and stability at large current density in alkaline electrolytes. Figure 2a shows that the overpotentials of $CuMo_6S_8$/Cu electrode are only 320 mV at 1000 mA cm$^{-2}$ and 334 mV at 2500 mA cm$^{-2}$, which are much smaller than that of $MoS_2$ (579 mV @1000 mA cm$^{-2}$) and Pt/C (474 mV @1000 mA cm$^{-2}$) electrodes. There is only 14 mV deviation of overpotential from 1000 mA cm$^{-2}$ to 2500 mA cm$^{-2}$, showing a small energy barrier for the $CuMo_6S_8$/Cu electrode operating at large current density. The overpotential of $CuMo_6S_8$/Cu electrode at 10 mA cm$^{-2}$ is 172 mV, and the intrinsic activity of $CuMo_6S_8$ is obtained by normalizing electrochemical surface area (ECSA) and mass of electrocatalyst

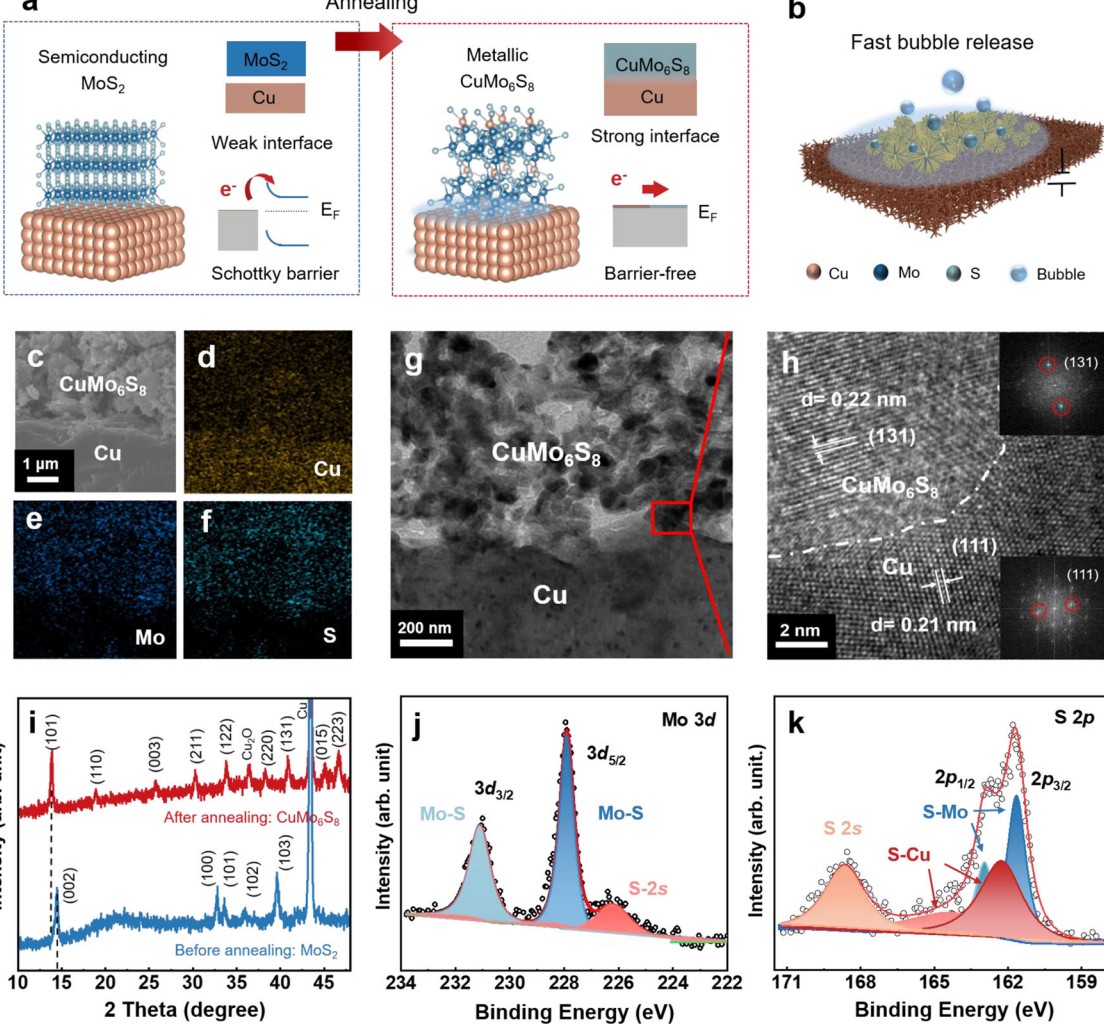

**Fig. 1 | Preparation and characterization of the Chevrel phase CuMo$_6$S$_8$/Cu electrode. a, b** A schematic showing the preparation process and macro-performance of CuMo$_6$S$_8$/Cu electrode. **c–f** Scanning electron microscopy (SEM) images and energy dispersive X-ray spectroscopy (EDS) mappings of a cross-section of the electrode. **g** Cross-sectional high-resolution transmission electron microscopy (HRTEM) image of the CuMo$_6$S$_8$/Cu electrode. **h** The zoom-in view taken from the red rectangular area in **g**, showing the interface between the CuMo$_6$S$_8$ layer and Cu substrate. The insets of **h** are the corresponding fast Fourier transform patterns for CuMo$_6$S$_8$ (top) and Cu (bottom). **i** X-ray diffraction patterns of the samples before and after annealing. **j, k** X-ray photoelectron spectroscopy spectra of Mo 3*d* (**j**) and S 2*p* (**k**) of CuMo$_6$S$_8$.

(Supplementary Figs. 11c–f). The performance of Pt/C electrode is also provided as a reference. Here the Pt/C electrode is prepared by dropping Pt/C ink on Cu foam, which shows similar performance compared with literature (Supplementary Table 2). To gain insights into performance at large current density, the slope of the polarization curve $\Delta\eta/\Delta\log|j|$ is implemented to evaluate the performance of the electrode at large current density (Fig. 2b)[40]. The $\Delta\eta/\Delta\log|j|$ ratios of the MoS$_2$ and Pt/C electrodes increase sharply with increasing current density, while that of the CuMo$_6$S$_8$/Cu electrode maintains a small value of 43 mV dec$^{-1}$, indicating its better mass transfer ability at large current density. In addition, the stability of the CuMo$_6$S$_8$/Cu electrode is tested by chronoamperometric (i-t) method at large current densities of 500, 1000, 2500 mA cm$^{-2}$ (Fig. 2c). The i-t curves show negligible degradation of performance within 300 h, especially the stability can be maintained at an ultra-large current density of 2500 mA cm$^{-2}$ for over 100 h, suggesting this electrode has an extremely stable structure.

For hydrogen evolution at large current density, interfacial charge transfer between catalyst and support significantly affects catalytic kinetics. Electrochemical impedance spectra show that the charge transfer resistance of CuMo$_6$S$_8$/Cu is twice and three times less than

that of the MoS$_2$ and Pt/C electrodes. (Supplementary Fig. 11a). This is because the interfacial Schottky barrier has been eliminated in CuMo$_6$S$_8$/Cu due to the transformation from semiconductor (MoS$_2$) to metal (CuMo$_6$S$_8$) (Fig. 5f). Besides, charge transfer resistance can be reduced further because we do not use Nafion binder compared with Pt/C electrode. Then, we compare with the reported state-of-art non-noble metal electrocatalysts operated at current density above 2000 mA cm$^{-2}$ from three aspects, that the $\Delta\eta/\Delta\log|j|$ ratio @ 1000 mA cm$^{-2}$–2000 mA cm$^{-2}$, $\eta$ @ 2000 mA cm$^{-2}$ and the largest tested $j$ of i-t test and its corresponding operation time (Fig. 2d, e, and Supplementary Table 3). It is seen that the $\Delta\eta/\Delta\log|j|$ ratio of CuMo$_6$S$_8$/Cu electrode is the smallest (32.5 mV dec$^{-1}$) and the $\eta$ @ 2000 mA cm$^{-2}$ of our work (324 mV) approaching the best one reported. Moreover, the tested $j$ of i-t test (2500 mA cm$^{-2}$) and its corresponding operation time (100 h) of our work are the largest and longest compared with reported literature. Besides, our work is also better than most other reported electrocatalysts operating at a current density of 500–1500 mA cm$^{-2}$ in terms of performance and stability. (Supplementary Fig. 12, Table 4). The above results show superior electrochemical performance and excellent stability of the CuMo$_6$S$_8$/Cu electrode at large current density, which is attributed to the

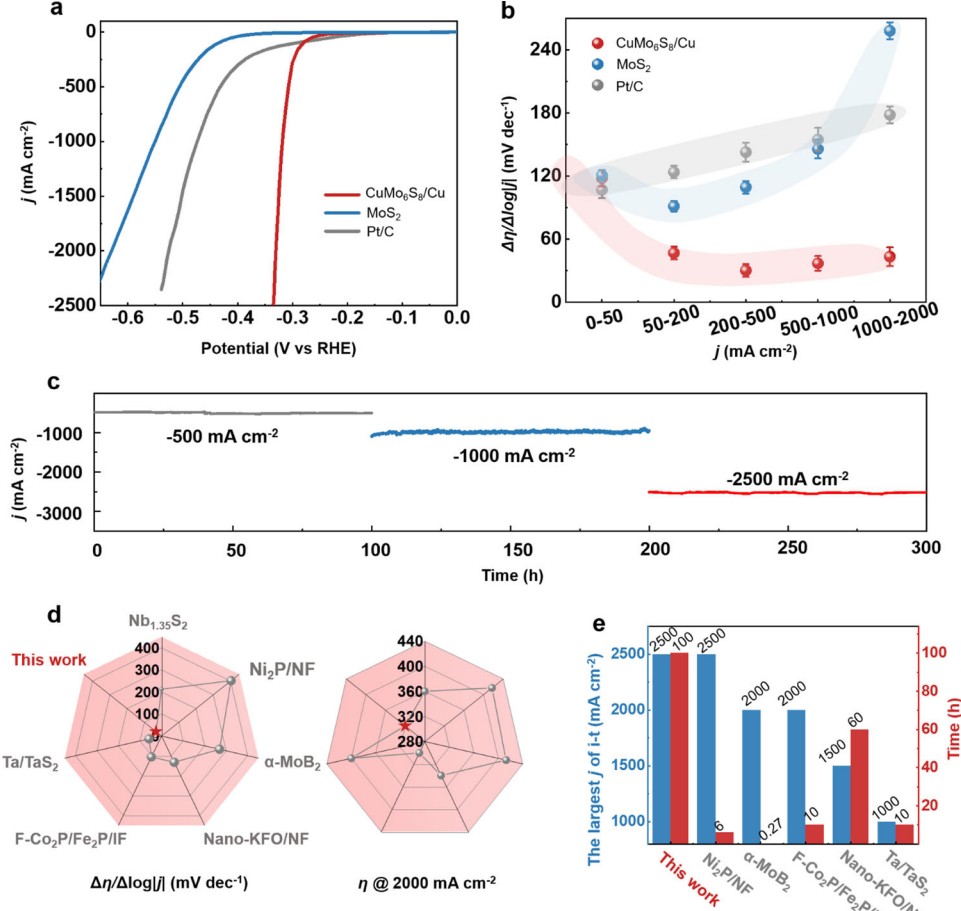

**Fig. 2 | HER performance at large current density. a** Polarization curves of three electrodes including CuMo$_6$S$_8$/Cu, MoS$_2$, and Pt/C. All tests are done in 1 M KOH at a scan rate of 1 mV s$^{-1}$ with 85% *iR* correction. **b** $\Delta\eta/\Delta\log|j|$ ratios of three electrodes at different current density ranges. **c** Chronoamperometric (i-t) curves of the CuMo$_6$S$_8$/Cu electrode at current densities of −500, −1000, −2500 mA cm$^{-2}$ over 300 h, where the current densities correspond to potentials of −0.43 V, −0.57 V, −0.94 V vs RHE without *iR* correction, respectively. **d**, **e** Comparisons of HER performance and stability of the Cu/CuMo$_6$S$_8$ electrode with non-noble metal electrocatalysts operated at a current density above 2000 mA cm$^{-2}$. From left to right: the $\Delta\eta/\Delta\log|j|$ ratio @1000 mA cm$^{-2}$−2000 mA cm$^{-2}$ and $\eta$ @2000 mA cm$^{-2}$ (**d**) and the largest tested *j* of i-t test and its corresponding time (**e**).

unimpeded interfacial charge transfer, abundant active sites and strong interfacial binding.

## In-situ total internal reflection imaging method to characterize mechanical stability of electrode

Then, we use an optical method of in-situ total internal reflection (TIR) imaging method[41,42] to get insightful information about the variation of the electrode activity. This electrode experiences 10,000 cyclic voltammetry (CV) cycles from 0 to −1500 mA/cm$^2$ to quantify the electrode stability at large current density. The schematic of our home-made TIR sensor system, its explosive views, and photos are shown in Supplementary Fig. 13. This system mainly consists of incident light module, detection module, and electrochemical cell module (see Methods for more details). The electrode is in close contact to a prism and the red light is incident at the interface between the prism and electrolyte at a critical TIR angle (Fig. 3a). Immediately when hydrogen evolution occurs, new microscopic interfaces will form between the electrode, H$_2$ bubbles and electrolyte in the evanescent layer (Fig. 3b). This working angle ($\Theta_{\text{working}}$) is set a little smaller than the critical TIR angle, which is 53.6° corresponding to the reflectivity of 0.75 in its sensitive region (Fig. 3c). The generated H$_2$ bubbles make the equivalent refractive index decrease, accompanied by a decrease in the intensity of the reflected light (Fig. 3c, Eq. (1)). The equivalent refractive index can be approximately expressed as:

$$n_{\text{equivalent}} = (1 - \upsilon) \times n_{\text{electrolyte}} + \upsilon \times n_{\text{bubble}} \tag{1}$$

Where $n_{\text{equivalent}}$ is the equivalent refractive index, $n_{\text{bubble}}$ and $n_{\text{electrolyte}}$ is the refractive index of H$_2$ bubble (1.000) and KOH electrolyte (1.409), respectively, $\upsilon$ is the volume concentration of H$_2$ bubbles in the evanescent layer. As shown in Fig. 3c, few bubbles generated ($\upsilon$ = 0.17%) in the evanescent layer can cause a large reduction of reflectivity. Therefore, the potential corresponding to mutation of reflective light intensity in one pixel can be defined as onset potential. It is worth noting that in this method, the onset potential results are obtained in the early instantaneous process, where the microbubbles are invisible due to their small sizes and very small amounts. This is like a "turn on" moment of the HER reaction. Note that the growth of microbubbles into big bubbles, as well as the subsequent diffusion and adhesion of bubbles to the prism all occur after the process of obtaining onset potential data, so it would not affect the test results (Supplementary Movie 1). As shown in Fig. 3d, the onset potentials of the Pt/C and CuMo$_6$S$_8$/Cu electrodes are −86 and −165 mV vs. reversible hydrogen electrode (RHE), which agrees well with the electrochemical test results and confirms the reliability of TIR imaging method. Here, the onset potential information of the pixels in the whole electrode area

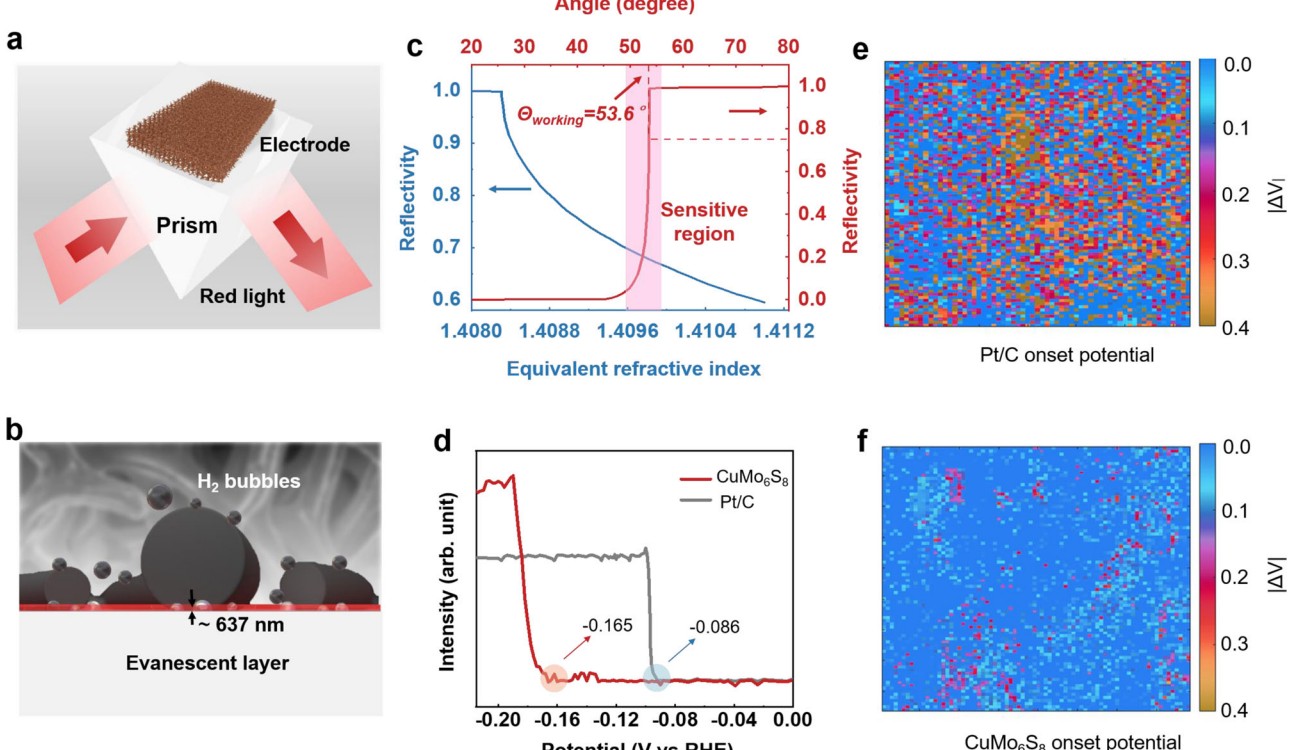

**Fig. 3 | In-situ total internal reflection imaging method to characterize electrode stability. a** Schematic of the macroscopic contact among the red light, prism and electrode. The incident red light is at the TIR angle. **b** The new-born microscopic interfaces between the electrode, $H_2$ bubbles, and electrolyte during HER. The evanescent layer is about 637 nm. **c** The relationships between incident angle, equivalent refractive index and reflectivity. **d** The relationships between onset potential of electrode and light intensity. **e, f** Difference value mappings of the onset potential of Pt/C and CuMo$_6$S$_8$/Cu electrode before and after 10,000 CV cycles.

before and after 10,000 CV cycles is in-situ collected and combined into mappings. The pristine views and onset potential mappings before and after 10,000 CV cycles are shown in Supplementary Figs. 14, 15. Mappings of difference values of onset potentials of the Pt/C and CuMo$_6$S$_8$/Cu electrodes are shown in Fig. 3e, f. Such difference values stem from electrocatalyst peel-off, and large difference values indicate severe peel-off. We find that the area of large difference values over 0.05 V occupies about 75% field of the Pt/C electrode. Inversely, the area of small difference values below 0.05 V accounts for about 82% for the CuMo$_6$S$_8$/Cu electrode (Supplementary Fig. 16b). This result show that the mechanical stability of the CuMo$_6$S$_8$/Cu electrode is much better than that of Pt/C and the interfacial binding through chemical covalent bonding is stronger than the commercial binder. In addition, the results above and the repeatability of the experiments are also verified by three parallel experiments for different CuMo$_6$S$_8$/Cu and Pt/C electrodes (Supplementary Figs. 17–19), as well as three replicate experiments on the same CuMo$_6$S$_8$/Cu electrode (Supplementary Fig. 20). Such stability of the CuMo$_6$S$_8$/Cu electrode is also confirmed by the XRD, polarization curves and ICP-OES of dissolved matters in the solution after i-t test, none of which shows noticeable change (Supplementary Fig. 21, Table 5, 6). Compared with traditional electrochemical methods, the TIR method not only gives the spatial information of activity distribution of the electrode at micrometer level, but also quantifies the peeling-off degree of electrocatalysts well, strongly confirming the mechanical stability of the CuMo$_6$S$_8$/Cu electrode. In addition, the TIR method also has unique advantages of large field of view, high detection sensitivity for hydrogen bubble formation (Supplementary Table 7, Note 3), easy operation and low requirements for equipment, compared with other optical imaging methods such as X-ray imaging.

## Strong interfacial binding force and weak electrocatalyst-gas bubble adhesion force

Here, we explain the reasons for excellent performance and mechanical stability of the CuMo$_6$S$_8$/Cu electrode from two aspects of electrocatalyst-support interfacial binding and electrocatalyst-gas bubble interfacial adhesion force. A simplified force analysis model of gas bubbles attached on the electrode is shown in Fig. 4a. The three forces, including the electrocatalyst-bubble interfacial adhesion force $F_a$, the electrocatalyst-support interfacial binding force $F_b$, and the gravity of electrocatalysts $G$ are in equilibrium at rest. The electrocatalysts can be adhesive on the support steadily by increasing $F_b$ and decreasing $F_a$. First we study the critical binding forces of the CuMo$_6$S$_8$ and Pt/C electrodes by micro scratch tester. The needle passes across the surface of the coating with an increasing normal force and the coating will begin to peel off at a certain normal force, which is defined as critical binding force. The tested results are confirmed by optical microscopy, friction-normal force curves and acoustic signals[43] (Fig. 4b, c and Supplementary Fig. 22). The CuMo$_6$S$_8$ peels off from the support at a load of 1.15 N while 0.58 N for Pt/C. The statistical data from multiple experiments confirm that the CuMo$_6$S$_8$/Cu electrode has stronger interfacial binding force than the Pt/C electrode.

Second, we in situ observe gas bubble evolution at 10 mA cm$^{-2}$ in a home-made cell by optical microscopy. We find that smaller gas bubbles release quickly from the CuMo$_6$S$_8$/Cu electrode, while bubbles tend to agglomerate into bigger ones on the Pt/C electrode (Fig. 4d, e and Supplementary Movie 2). The insets of Fig. 4d, e show the contact angles of CuMo$_6$S$_8$ (~0°) and Pt/C (81.3°) electrodes, manifesting the CuMo$_6$S$_8$/Cu electrode is more hydrophilic than the Pt/C electrode. Figure 4f shows evolution rate of gas bubbles on the CuMo$_6$S$_8$/Cu electrode is 213.9 mm$^{-2}$ s$^{-1}$, much higher than that on the Pt/C (30.6 mm$^{-2}$ s$^{-1}$), but the bubble diameter is smaller (0.12 mm *vs.*

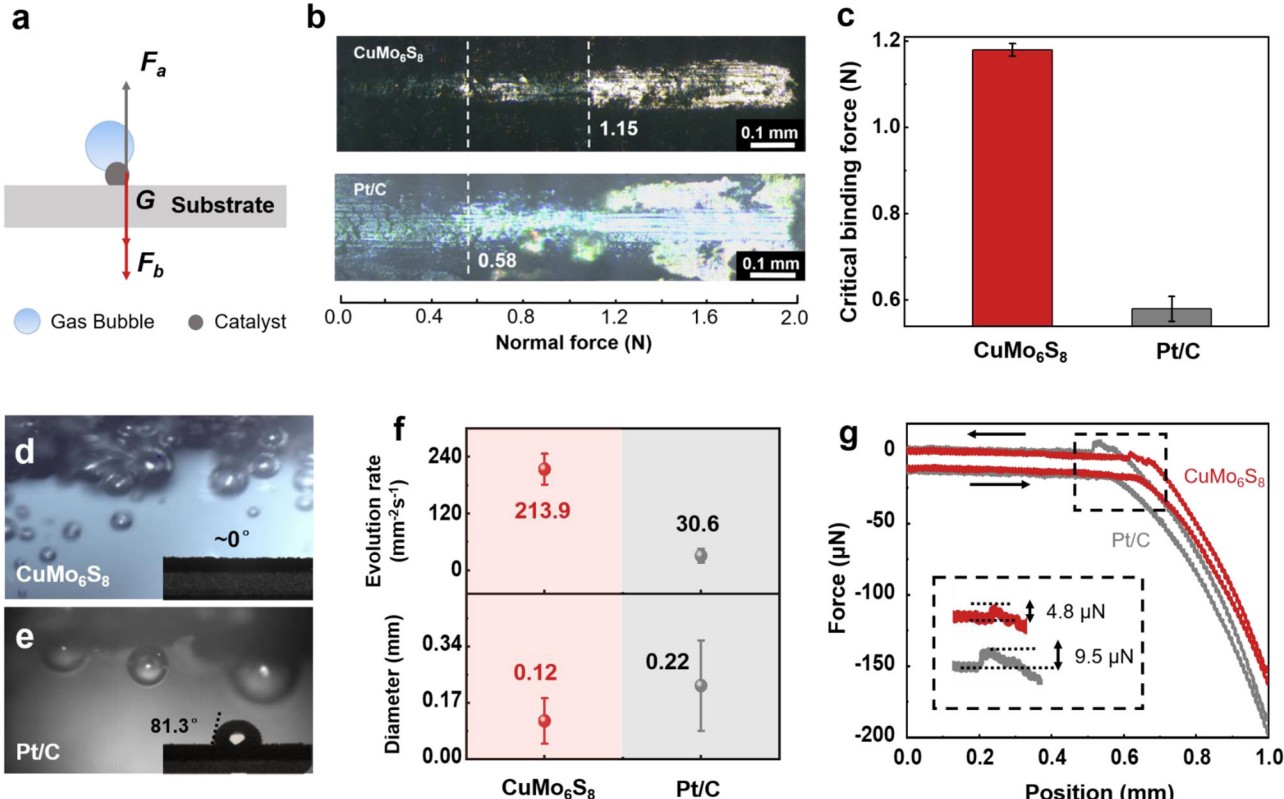

**Fig. 4 | Strong interfacial binding force and small electrocatalyst-gas bubble interfacial adhesion force. a** Force analysis model of gas bubbles attached on the electrode. $F_a$, $F_b$, and $G$ represent the electrocatalyst-bubble interfacial adhesion force, the electrocatalyst-support interfacial binding force, and the gravity of electrocatalysts, respectively. **b** The photos of micro-scratches of $CuMo_6S_8$ (top), Pt/C (bottom) catalyst layers adhesive on Cu foils. The dotted lines represent the critical adhesive forces corresponding catalysts. **c** The statistical data of critical adhesive forces of $CuMo_6S_8$ and Pt/C. The error bars represent the statistical distribution of three samples. The photos of gas bubbles generated on **d** $CuMo_6S_8$ and **e** Pt/C electrodes at 10 mA cm$^{-2}$. The insets show the corresponding contact angles of -0° and 81.3°. **f** Evolution rate and diameter of gas bubbles of $CuMo_6S_8$ and Pt/C electrodes. The error bars represent the distribution of statistical values. **g** Electrocatalyst-bubble interfacial adhesion force of $CuMo_6S_8$/Cu and Pt/C electrodes.

0.22 mm). Here we can combine theoretical derivation and experimental data to quantitatively analyze the $F_a$. Whether the bubbles can be attached to the electrocatalysts depends on the change of the interfacial energy before and after attachment, defined as ΔW, as expressed by Eq. (2):

$$\Delta W = \sigma_{GL}(\cos\theta - 1), \tag{2}$$

where $\sigma_{GL}$ is the interfacial energy between gas and liquid phases, $\theta$ is the equilibrium contact angle. Detailed derivations are shown in Supplementary Note 1. It is known from Eq. (2) when the $\theta$ is near zero degrees, that the electrocatalysts are superhydrophilic, gas bubbles are not inclined to attach to the electrode, which means the $F_a$ is not large enough to cause the electrocatalysts to peeling-off. This analysis has been confirmed by our electrocatalyst-bubble interfacial adhesion force test (Fig. 4g), which shows the $F_a$ of the $CuMo_6S_8$/Cu electrode (4.8 μN) is much smaller than that of the Pt/C electrode (9.5 μN). Moreover, HER performance at large current density is largely affected by gas bubble detachment. The growth and accumulation of gas bubbles on the catalyst surface would produce micro-convection, impede ion transport and block active site, causing extra energy consumption namely transport overpotential ($\eta_{trans}$)[44,45]. The results of $\eta_{trans}$ at different current densities are plotted in Supplementary Fig. 23 and detailed derivations are shown in Supplementary Note 2. Obviously, the $\eta_{trans}$ and its increasing trend of the $CuMo_6S_8$/Cu electrode at large current density is much smaller than that of the Pt/C. This is because the $CuMo_6S_8$/Cu electrode has a small catalyst-bubble interfacial adhesion force, thus exhibits faster bubble evolution

kinetics. The above results jointly illustrates strong interfacial binding force and small electrocatalyst-bubble interfacial adhesion force are responsible for mechanical stability and performance of the $CuMo_6S_8$/Cu electrode at large current density.

### Active sites and metallic property of $CuMo_6S_8$

The effect of chalcogen electronegativity of Chevrel phase chalcogenides ($Mo_6X_8$; X= S, Se, Te) on HER activity has been investigated previously[46]. Here we further study the relationship between the coordination of sulfur atoms on main crystal facets of $CuMo_6S_8$ and HER activity. Figure 5a–e show the HER active sites on three main facets of $CuMo_6S_8$ and corresponding free energy diagrams. The adsorption free energy of H* ($\Delta G_{H*}$) is a reasonable descriptor of HER activity. The ideal HER electrocatalyst should have a moderate $\Delta G_{H*}$ which is close to 0 eV. Our calculation results show that pristine $MoS_2$ has a bad HER performance with a very high $\Delta G_{H*}$, which is consistent with experimental results. S atoms with coordinate numbers of 3, 2 and 3 in $CuMo_6S_8$ (001), (101) and (110) facets have the $\Delta G_{H*}$ of −0.084 eV, 0.200 eV and −0.221 eV, respectively (Fig. 5e). All of them are excellent active sites for HER, approaching the $\Delta G_{H*}$ of Pt. $\Delta G_{H*}$ of other exposed atoms in these facets are also tested, as shown in Supplementary Fig. 24. It is seen that S atoms with lower coordinate numbers usually show lower $\Delta G_{H*}$ than those with higher coordinate numbers in each facet. To reveal the causal relationship between the coordinate number of S atom and $\Delta G_{H*}$, the Bader effective charges of S atoms with different coordinate numbers are calculated and shown in Supplementary Table 8. The S atom with larger coordinate number shows more negative Bader effective charge. Since the S atom is more

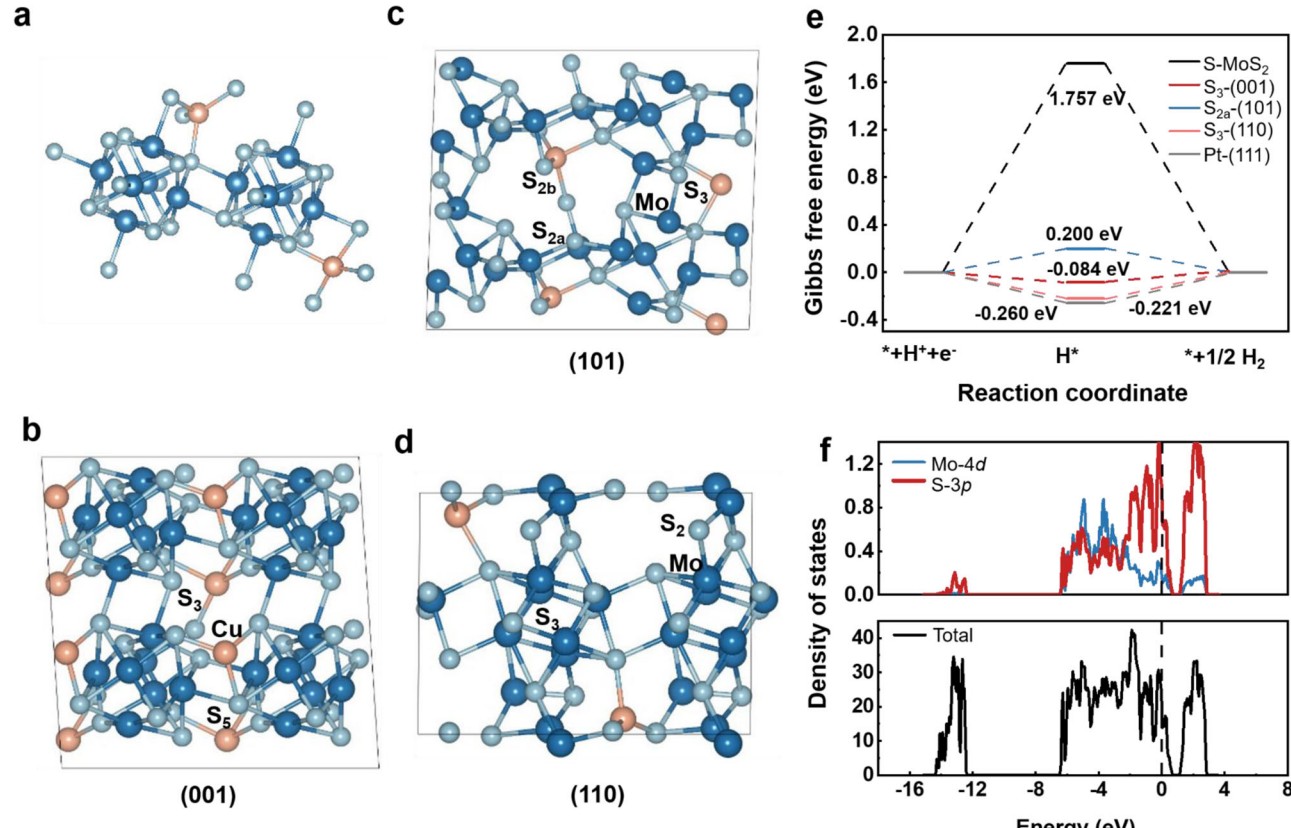

**Fig. 5 | Active sites and metallic property of CuMo₆S₈. a** Atomic structure of CuMo₆S₈. **b–d** Exposed (001), (101), (110) facets. Orange, dark blue and cyan spheres represent the copper, molybdenum and sulfur atoms, respectively. The marks show the main active sites in different crystal facets and corresponding subscripts represent the coordination number. **e** Gibbs free energy ($\Delta G_{H^*}$) variations for HER on best active sites of three facets of CuMo₆S₈, sulfur site of MoS₂ and Pt (111). **f** Density of states (DOS) and partial density of states (PDOS) of CuMo₆S₈.

electronegative than H in S-H bond, a larger coordinate number of S (already has more negative Bader effective charge) should show a weaker S-H bond strength, i.e., a higher $\Delta G_{H^*}$. From this perspective, the coordinate number of S atom is a descriptor of HER activity for each facet considered in this study. Noted that this rule only works for S atoms on the same facet but not on different facets. There are plenty of exposed S atoms with coordinate numbers of 2 or 3, which may lead to the excellent HER performance of CuMo₆S₈. We also investigate the conductivity of CuMo₆S₈ and the density of states (DOS) are shown in Fig. 5f. Total DOS clearly shows that CuMo₆S₈ is a metal as the valence bands are partially filled. Partial density of states (PDOS) indicates that the DOS on Fermi energy level mostly originate from Mo-4$d$ and S-3$p$ states. In comparison, 2H-phase MoS₂ is a semiconductor as the valence bands are completely filled with electrons and separated from the conduction bands with a band gap (Supplementary Fig. 25). Therefore, a higher electron transfer efficiency in metallic CuMo₆S₈ than that on the semiconductor of 2H-phase MoS₂ is expected. Taken together, the above results suggest that CuMo₆S₈ is a metal with high intrinsic HER activity.

## Discussion

In summary, we construct a Chevrel phase CuMo₆S₈/Cu electrode with superior mechanical stability and HER performance by dual interfacial engineering. The Chevrel phase CuMo₆S₈ originates from MoS₂ and binds with support firmly by chemical bonding. It features a strong electrocatalyst-support binding force and a weak electrocatalyst-bubble adhesion force. As a result, the CuMo₆S₈/Cu electrode achieves excellent performance with a small overpotential of 334 mV@ 2500 mA cm⁻² and operates at 2500 mA cm⁻² stably over 100 h. The mechanical stability of the CuMo₆S₈/Cu electrode at large current

density is quantitatively described by in situ TIR imaging method with a spatial resolution of micrometer, showing peeling-off degree of electrocatalysts on CuMo₆S₈/Cu electrode is nearly four times smaller than that of the Pt/C electrode. Furthermore, mechanical tests clarify the influences of interfacial forces on mechanical stability and HER performance at large current density. Theoretical calculations show that CuMo₆S₈ has metallic property and high activity, effectively promoting the interfacial electron transport and hydrogen evolution kinetics. The dual interfacial engineering deepens the understanding about the effect of interfacial property on electrode stability and electrochemical performance, which is expected to be used in other gas-involving electrochemical reactions toward constructing stable and highly efficient electrodes.

## Methods
### Material preparations

All chemicals were used as received without further purification. First, we used the intermediate-assisted grinding exfoliation to produce 2D MoS₂, where bulk MoS₂ (Aladdin Chemical Reagent Co., 99.5%, diameter <2 μm, 10 g) and Fe powders (serve as force intermediates, Aladdin Chemical Reagent Co., 98%, 400 mesh, 0.01 g) were added together into a mortar grinder (Retsch RM 200, Germany) and ground for 9 h. The sample was further centrifuged with 10,000 rpm, followed by pickling with 1 M HCl and washing with deionized water. Then, the obtained 2D MoS₂ were dispersed in ethanol and treated by ultrasonic for 10 min. The Cu foams were first treated by CV cycles in 0.5 M H₂SO₄ to improve the surface roughness. After that, the MoS₂ ink was loaded on the treated Cu foams by drop casting or spray coating and the loading amount was 10 mg cm⁻². Finally, the dried MoS₂/Cu electrodes were annealed under the Ar

(200 sccm) and $H_2$ (10 sccm) at 750 °C for 2 h to prepare the CuMo$_6$S$_8$/Cu electrodes. The Pt/C electrode was obtained by dropping the commercial Pt/C ink (20 wt%) on the Cu foams. The Pt/C ink consisted of 2 mg Pt/C powder (Macklin Biochemical Co., 20%), 0.4 mL iso-propanol, 1.5 mL deionized water and 0.1 mL Nafion binder (Dupont Co., 5 wt%). The loading amount of Pt/C was 0.5 mg cm$^{-2}$. The specification of pure Pt foil electrode was $1 \times 1$ cm$^2$.

## Material characterizations

Phase and crystal structure of samples were characterized by XRD (Cu $K_\alpha$ radiation, $\lambda = 1.54$ Å, Bruker D8 Advance, Germany). The surface morphology and elementary composition were characterized by SEM (Hitachi SU8010, 20 kV). HRTEM images were obtained by using an electron acceleration voltage of 300 kV (FEI Tecnai F30, USA). The thicknesses of 2D MoS$_2$ nanoflakes were measured by AFM (Oxford Asylum Research, UK). Raman spectra were collected by using a 532 nm laser as excitation light (Horiba LabRAM HR Evolution, Japan). Chemical analysis was performed by high-resolution XPS (monochromatic Al K$\alpha$ X-rays, Thermo Fisher ESCALAB 250Xi, USA). The mechanical property of the electrodes was characterized by micro scratch tester (Anton Paar, UNHT, Austria). The contact angle was tested by contact angle meter (KRUSS DSA30, Germany). The electrocatalyst-bubble adhesion force was tested by high-sensitivity micro-electrochemical balance (DataPhysics DCAT21, Germany).

## Electrochemical measurements

All electrochemical measurements were conducted by using an electrochemical workstation (Biologic Co., VMP300, France). In all tests, a solution of 1 M KOH electrolytes was used. The Hg/HgO electrode and graphite rod were used as reference and counter electrodes in a standard three-electrode cell. The working and reference electrodes were fixed as close as possible to reduce the solution resistance to be below 0.3 Ω and the wetted area of all tested electrodes was all tailored to be 1 cm$^2$ for fair comparison among samples. The following equation was used to convert the applied potential from $vs$ Hg/HgO to $vs$ RHE:

$$E_{vs\,RHE} = E_{vs\,Hg/HgO} + 0.059 * pH + 0.098 \tag{3}$$

Before tests, the electrolyte was purged with Ar gas for 10 min to exclude oxygen. The linear sweep voltammetry (LSV) was performed at a scan rate of 1 mV s$^{-1}$ with 85% $iR$ correction. The cyclic voltammetry (CV) was applied at a scan rate of 50 mV s$^{-1}$. The chronoamperometric (i-t) was applied at a current density of 500, 1000, and 2500 mA cm$^{-2}$ for the 300 h stability tests. Electrochemical impedance spectroscopy was performed at the potential corresponding to a current density of 10 mA cm$^{-2}$ with frequencies from 1 MHz to 0.1 Hz. Electrochemical active surface areas (ECSA) of samples were obtained by dividing the electric double layer capacitance by the specific capacitance. Since the specific capacitances of MoS$_2$-based materials are between 20–60 µF cm$^{-2}$ [1-4], so we chose 40 µF cm$^{-2}$ to calculate the ECSA. Besides, most Pt-based materials reported showed a specific capacitance between 28 and 60 µF cm$^{-2}$ in alkaline electrolyte [5-8], so we chose 40 µF cm$^{-2}$ as our specific capacitance for Pt/C and Pt foil.

## TIR measurements

The TIR sensor system (Supplementary Fig. 13a) consisted of three parts, including incident light module, detection module, and electrochemical cell module. In the incident light module, the light emitted by a LED (LR W5AP, Osram, Germany, center wavelength 633 nm, electric power 5 W) was collected by a ×25 objective lens (GCO-2104, Daheng Optics, China) and focused on a home-made 0.3 mm pinhole. The light from the pinhole was collimated by an achromatic convex lens (GCL-010650, Daheng Optics, China, focal length 30 mm), passed a bandpass filter (FL632.8-10, Thorlabs, USA, center wavelength 632.8 nm, bandwidth 10 nm) and a linear polarizer (GCL-050003,

Daheng Optics, China, extinction ratio 500:1), then radiated the electrochemical cell module. The light reflected by the electrochemical cell module was collected and digitized by the detection module. The detection module was composed of an imaging lens (HF-5MPB50, YVISION, China, focal length 50 mm, with adapter rings) and a CCD camera (Retiga R3, Qimaging, Canada, $1920 \times 1460$ pixels, $4.54 \times 4.54$ µm$^2$ pixel size, thermoelectric cooling to $-20$ °C). The explosive view of the electrochemical cell module was shown in Supplementary Fig. 13b. A Ti wire was inserted from the side of a Cu foam as the working electrode, another Ti wire and an Hg/HgO electrode were the counter electrode and the reference electrode. The photos of the entire TIR system and the electrochemical cell module were shown in Supplementary Fig. 13c–e.

LSV and CV test were performed on the electrochemical module. Before measurements, 1.0 M KOH solution with a volume of 10 mL was injected into the electrolytic cell, and the working electrode was soaked in it followed by vacuuming. Then, the electrochemical cell module was installed in the TIR system and modulated to achieve a working angle of 53.6°. After that, high-resolution images were achieved by adjusting the imaging lens of the detection module. Before 10,000 CV, the first LSV test was initially performed with a scan rate of 1 mV s$^{-1}$ at the potential window of 0 to $-0.695$ V vs RHE without $iR$ correction. Meanwhile, TIR test was also performed with an exposure time of 6 ms and a collection interval of 2 s. Therefore, the electrochemical information and images can be acquired simultaneously to obtain the initial onset potential mapping. After that, the CV test of 10,000 cycles (0 to $-1500$ mA cm$^{-2}$, w/o $iR$ correction, Supplementary Fig. 16a) was performed with a scan rate of 50 mV s$^{-1}$ followed by second vacuuming to exclude accumulated bubbles, and TIR test to acquire the onset potential mapping after 10,000 CV. Post-processing and statistics were performed to compare the numerical difference before and after 10,000 CV.

## Computational methods

All the density functional theory (DFT) calculations were performed via the Vienna Ab initio Simulation package [10-14], and the projector-augmented plane wave pseudopotentials were used for the elements involved [12]. The generalized gradient approximation of Perdew, Burke, and Ernzerhof was used to treat the exchange correlation between electrons [15]. CuMo$_6$S$_8$ (110), (001) and (101) surfaces were investigated for HER and the bottom layers kept fixed during the calculation. A vacuum region of greater than 15 Å was added along the direction normal to the slab plane to avoid the interaction between periodic supercells. The electron wave function was expanded in plane waves and a cutoff energy of 500 eV was chosen. The Monkhorst-Pack meshes of ($3 \times 3 \times 1$) were adopted for the Brillouin zone of the slabs and primitive cell [16]. The convergence in the energy and force were set to be $10^{-4}$ eV and 0.01 eV/Å, respectively.

The free energies of H$_2$O (l) and H$_2$ (g) were used as references when calculating the free energies of reaction intermediates. The adsorption energy for reaction intermediate was calculated as follows [17]:

$$\Delta G = \Delta E_{Total} + \Delta E_{ZEP} - T\Delta S + \Delta G_s \tag{4}$$

where $\Delta E_{Total}$ is the calculated adsorption total energy by DFT, $\Delta E_{ZPE}$ is zero-point energy, $\Delta S$ is entropy, and $\Delta G_s$ is solvation energy [18,19]. The calculated HER electrochemical potential can be obtained as follows:

$$U_L = Mini[-\Delta G_i]/ne \tag{5}$$

where $n$ is the number of electrons transferred for each electrochemical step, and $e$ is the elementary charge. Here, the $n$ is set to 1 for the one-electron transfer step. The meaning of the r.h.s. of the above equation is to select the smallest $[-\Delta G_i]$ among the HER elementary steps.

## Data availability

All data are available from the authors upon reasonable request. Source data are provided with this paper.

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

## Acknowledgements

We acknowledge financial support from the National Science Fund for Distinguished Young Scholars (No. 52125309), the National Natural Science Foundation of China (Nos. 52188101, 51920105002, 51991343, 51991340, and 61975089), the Guangdong Innovative and Entrepreneurial Research Team Program (No. 2017ZT07C341), the Shenzhen Basic Research Project (Nos. JCYJ20200109144620815 and JCYJ20200109144616617), Science and Technology Research Program of Shenzhen City (JCYJ20200109110606054), and the National Key Research and Development Program of China (Nos. 2017YFA0700103 and 2018YFA0704502).

## Author contributions

H.L., Y.L., and B.L. conceived the idea. H.L. synthesized the materials and performed most of the materials characterization and electrochemical tests. Z.C. and L.L. performed the TIR experiments and analysis. R.X. and G.C. performed DFT calculations and analysis. Z.G. and X.G. performed the bubble adhesion force tests and analysis. Q.Y., Y.L., Z.Z., F.Y., X.K., S.G., S.L., R.X., and Z.C. participated in the discussion. B.L. supervised the project and directed the research. H.L., X.G., L.L., G.C., and B.L. interpreted the results. H.L. and B.L. wrote the manuscript with feedback from the other authors.

## Competing interests

The authors declare no competing interests.
