## [Peer Review File · Nature Communications]

Title: Dual interfacial engineering towards superior and stable hydrogen evolution at 2500 mA cm⁻²REVIEWER COMMENTS

Reviewer #1 (Remarks to the Author):

The authors reported that a mechanically stable all-metal shevral phase of CuMo6S8 on Cu mesh electrode for the hydrogen evolution. Their HER electrocatalysts with dual interfacial engineering comes from 2H-phase MoS2 on Cu foam by high temperature annealing showing stable HER even at a large current density. The reviewer shows couple of concerns; 1) the interfaces in between annealed 2H MoS2 and then Shevral phase of CuMo6S8 with Cu seems not secured electrically and structurally. Looking in Fig. 1C. porous CuMo6S8 on Cu looks not stable. Please provide more microstructurally evidences. and How come to detect the XPS peaks from Cu-S chemical binding at the interfaces to the near surfaces? 2) The authors should provide the thickness of whole electrocatalysts of CuMo6S8/Cu. otherwise please provide surface roughness factor for the fair comparison of the large current density. Overall, the reviewer cannot recommend the manuscript can be published as present. There are quite a few studies on Shevral phase of CuMo6S8 via reaction in between MoS2 and Cu at high temperature. Novelty of the manuscript is also lack, otherwise state more unique point of their studies. The manuscript should be improved further by major revision or submit more specialized journals.

Reviewer #2 (Remarks to the Author):

This manuscript presents a novel synthesis approach to generate a binder-free CuMo6S8/Cu electrode and evaluates their hydrogen evolution activity under alkaline conditions. The electrodes exhibit high current densities at relatively low overpotentials as well as high mechanic stability, which was evaluated through In-situ total internal reflection imaging. The electrocatalyst-support interfacial binding and electrocatalyst-gas bubble interfacial adhesion force of the electrodes are evaluated using a force analysis model, which is experimentally validated through micro scratch testing and optical microscopy. The authors also include the adsorption free energy calculations for various CuMo6S8 facets. It is observed that facets with undercoordinated sulfur atoms have more favorable H adsorption energies. Badger analysis indicates that this is due to a higher electronegativity difference between S and H, leading to stronger S-H bonds. Lastly, density of states and partial density of states calculations of CuMo6S8 are used to elucidate the metallic nature of CuMo6S8, which is expected to have a higher electron transfer efficiency than that of semiconducting 2H-MoS2. Based on the novelty of the electrode generation, high current densities, and the extensive characterization of electrocatalyst support/ bubble adhesion force I consider that Nature Communications is an appropriate platform to publish the manuscript. However, the following suggestions will improve the clarity and reproducibility of the work.

1. The use of "fast hydrogen evolution" instead of "fast bubble evolution" in the 4th sentence of the abstract seems more appropriate since the reaction under study is currently not mentioned in the abstract at all. The authors should also emphasize that the electrocatalyst are evaluated under basic conditions in the abstract. This detail increases the significance of the work since Chevrel Phases have been extensively evaluated as HER catalyst in acidic media, but very rarely in basic media.

2. The significance of the TEM image shown in Fig. 1h will increase if the reader could identify the (131) and (110) planes in the XRD shown in Fig. 1i. The authors are encouraged to index the index pattern to facilitate this comparison.
3. To facilitate the comparison between MoS₂ and CuMo₆S₈ of Fig. 1i the authors are encouraged to provide a zoom-in version of the peak between 10-20 2θ in the SI.
4. References for the overpotentials and current densities included for MoS₂ and Pt/C electrodes in the following sentence should be included: "Fig. 2a shows that the overpotentials of CuMo₆S₈ /Cu electrode are only 320 mV at 1000 mA cm⁻² and 334mV at 2500 mA cm⁻², which are much smaller than those of MoS₂ (579 mV @1000mA cm⁻²) and Pt/C (474 mV @1000 mA cm⁻²) electrodes."
5. The authors should include the specific capacitance used to calculate the ECSA of the electrodes evaluated as well as the equation used to convert the applied potential from Hg/HgO to RHE.
6. Since the overpotential at 10ma/cm² has been extensively used as a figure of merit for HER catalyst, the authors should include it in the manuscript or SI to facilitate the comparison with other HER catalysts besides the ones included in table 2 of the SI.
7. The authors should include the potential required to achieve the current densities in figure 2c.
8. The H-S bond strength argument obtained from the badger charge analysis has been previously elucidated in Chevrel Phase chalcogenides and should be appropriately cited (<https://doi.org/10.1021/acsami.0c07207>).

Reviewer #3 (Remarks to the Author):

In this manuscript, a mechanically stable Chevral phase CuMo₆S₈/Cu electrode for electrocatalytic hydrogen evolution reaction is reported. The authors claimed that the dual interfacial engineering strategy enables resistance against peeling-off of electrocatalysts and improved reaction kinetics at large current density, resulting in better HER performance. The results are potentially interesting to readers in the field. However, there are still some technical issues that require clarification.

In-situ total internal reflection imaging method was used to study the stability difference between Pt/C and CuMo₆S₈/Cu electrodes before and after 10,000 cyclic voltammetric scans. Due to the evanescent field imaging requirement, the electrode surface must be in close contact with the prism. Considering that HER is a three-phase reaction, the diffusion of the hydrogen gas and the generation of bubbles have pivotal role in the process. In the experiment, the prism has great influence on the diffusion of hydrogen gas. The weak bubble adhesion at the electrode surface seems to be one of the key selling points of the

research. However, bubble adhesion on the prism was not well discussed, which might be the major contributing cause in observed results. These two factors, blocked hydrogen diffusion and surface bubbles on the prism, could lead to apparent deviation of testing condition from the real application.

The imaging technique could still provide some insight of the structure and activity change of the electrode surface. Unlike the previous carbon fiber experiment, the structures of which can be well resolved, the surface of the electrodes are almost homogenous and very random. Information from a single or a few in-situ experiments is not very convincing. Multiple parallel experiments and statistical analysis of the results would be sufficient.

The spatial resolution of the imaging technique is at micrometer scale, which is in the same range of other well-developed in situ techniques. Operando X-ray imaging, which offers similar resolution and minimum interference to the testing cell, could be a better alternative to the current method.

Reviewer #4 (Remarks to the Author):

Comments: In the manuscript, the author designed a stable and highly active CuMo₆S₈/Cu electrode by in-situ reaction between MoS₂ and Cu. The strong binding at electrocatalyst-support interface while weak adhesion at electrocatalyst-bubble interface made CuMo₆S₈ reach large current density and highly stable in Hydrogen Evolution Reaction (HER). However, the catalysts, the device performance and the claimed novel investigation method have been reported. Furthermore, the mechanism and intrinsic activity is not clear and in-depth in this research. Considering the novelty of this manuscript, I recommend reconsider after major revision,

Some points for the author:

1. In supplementary Fig. 2, the morphology of CuMo₆S₈ is not uniform, and there is also a -ribbon-like structure, which need to be confirmed.
2. The particle size of Pt/C is not given in the manuscript, and the particle size of CuMo₆S₈ and MoS₂ is different, which may also affect the performance comparison.
3. The XPS results showed the presence of S-Mo and S-Cu bonds. Is it possible that Mo-O bonds or S-O bonds are formed at the interface due to the presence of Cu₂O according to the XRD results?
4. The loading mass of the catalyst is an important factor affecting the HER activity. Whether the loading mass of CuMo₆S₈/Cu and Pt/C is the same, it is recommended to supplement the mass activity of the catalysts.
5. The author the HER active sites on three main facets of CuMo₆S₈, showing the CuMo₆S₈ (001), (101) and (110) facets are excellent active sites for HER, which is the best active site?
6. The author studied the effect of facets and coordinate number of S atom to performance, which is the most essential factor leading to the good performance of the CuMo₆S₈, what's the relationship between them?

Response to Reviewer #1

Comment. The authors reported that a mechanically stable all-metal Chevral phase of CuMo_6S_8 on Cu mesh electrode for the hydrogen evolution. Their HER electrocatalysts with dual interfacial engineering comes from 2H-phase MoS_2 on Cu foam by high temperature annealing showing stable HER even at a large current density. The reviewer shows couple of concerns:

Response. Thank you very much for your recommendations. We appreciate the reviewer by commenting that our electrocatalysts show stable HER at large current density, which is indeed the main point of this work.

Comment 1. The interfaces in between annealed 2H MoS_2 and then Chevral phase of CuMo_6S_8 with Cu seems not secured electrically and structurally. Looking in Fig. 1C, porous CuMo_6S_8 on Cu looks not stable. Please provide more microstructurally evidences, and how come to detect the XPS peaks from Cu-S chemical binding at the interfaces to the near surfaces?

Response 1. Thank you very much for pointing out this important point. The Fig. 1c in previous version is a low-magnification SEM image showing that CuMo_6S_8 has a porous structure, which is not sufficient to examine the interface as you mentioned. Inspired by your comments, we performed additional experiments to provide more microstructural evidences. **First**, we conducted the cross sectional SEM imaging to examine the morphology of CuMo_6S_8 on Cu foam. There is a clear and well-defined interface between Cu foam and the CuMo_6S_8 layer (Fig. R1a). We also find many microscopic pores existing in the CuMo_6S_8 layer, suggesting its porous feature (Fig. R1b). Such a porous structure leads to superhydrophilicity of the electrodes, which can avoid the large bubble/catalyst interface adhesion caused by the massive accumulation of hydrogen bubbles and further prevent the falling-off of catalyst. This is an important reason why our electrode shows decent performance at large current density, as has been discussed in detail in original manuscript. **Second**, we performed more high magnification SEM imaging and EDS element mappings of cross section of the

electrode (Fig. R2). We find that the Cu atoms diffuse from the Cu foam to MoS₂, and react with MoS₂ to form Chevral phase CuMo₆S₈ during thermal annealing. Such high temperature annealing usually leads to a strong interface. **Third**, most importantly, we conducted many additional HRTEM characterization directly at the interface between Cu and CuMo₆S₈ (Fig. R3). FIB cutting was used to expose the interface for HRTEM imaging. The zoom-in view of Fig. R3b shows the interface between the CuMo₆S₈ and Cu substrate. The lattice spacings of 0.22 nm and 0.21 nm correspond to (131) and (111) planes of CuMo₆S₈ and Cu, respectively. The insets are the corresponding FFT patterns, which also show the crystalline nature of CuMo₆S₈ and Cu. The dotted line indicates the interface between them, which is very smooth, indicating a close contact between CuMo₆S₈ and Cu with well-defined robust interface. Taken together, the above three additional sets of experiments provide microstructural evidence about the interface between CuMo₆S₈ and Cu.

Regarding the XPS of Cu-S bonding you mentioned, it maybe that we did not make it clear in the original manuscript. Here we make supplementary explanation. On one hand, the XPS information shown in manuscript is mainly from the CuMo₆S₈ layer that cover on the surface of Cu foam, not from the interface between Cu and CuMo₆S₈. On the other hand, we think it is hard to distinguish the XPS signals from the interface and near surface due to two reasons. **First**, the spatial resolution of XPS is limited to spot size of X-ray (~100 μm in our facility). Considering that both CuMo₆S₈ and Cu foam itself are rough over 100 μm size level, it is difficult for XPS covering 100 μm size level to get the information only from the interface while not the near surface information of the electrode. **Second**, we also tried to get XPS depth signals of interface by ion etching. However, the rough and porous CuMo₆S₈ layer cannot be uniformly thinned, which means that besides the XPS signals from the interface, signals from the surface CuMo₆S₈ residue and internal Cu substrate may be also included. Therefore, based on the discussion above, we think maybe it is not suitable for XPS to distinguish the chemical information from interface and near surface. We have added a sentence on page 5 of revised manuscript to make this point clear.

Changes to the revised manuscript. Some sentences were added in Page 5 of

revised manuscript as follows. “X-ray photoelectron spectroscopy (XPS) survey spectrum is also implemented to detect the chemical bonding of the CuMo_6S_8 layer that cover the surface of Cu foam, and shows that there are five elements (Cu, Mo, S, C and O) presented (Supplementary Fig. 7)”. On Page 6, Fig. R2a-d were used as new Fig. 1c-f, and Fig. R3a, b were used as new Fig. 1g-h. Corresponding captions and descriptions were also revised.

Figure R1. (a) Cross sectional SEM image of the $\text{CuMo}_6\text{S}_8/\text{Cu}$ electrode and (b) enlarged SEM image taken from the red rectangular area in (a), showing that CuMo_6S_8 possesses porous structure. The figure R1b was added as figure S4a in the revised SI.

Figure R2. (a) Cross sectional SEM image and (b-e) corresponding EDS mappings of $\text{CuMo}_6\text{S}_8/\text{Cu}$ electrode. These figures were added as Figs. 1c-f in revised manuscript.

Figure R3. (a) Cross sectional high resolution transmission electron microscopy (HRTEM) image of the $\text{CuMo}_6\text{S}_8/\text{Cu}$ electrode. (b) The zoom-in view taken from the red rectangular area in (a), showing the interface between the CuMo_6S_8 layer and Cu substrate. The insets of (b) are the corresponding fast Fourier transform patterns for CuMo_6S_8 (top) and Cu (bottom). These figures were added as Figs. 1g-h in revised manuscript.

Comment 2. The authors should provide the thickness of whole electrocatalysts of $\text{CuMo}_6\text{S}_8/\text{Cu}$. Otherwise please provide surface roughness factor for the fair comparison of the large current density.

Response 2. Thank you very much for your instructive suggestions. We have followed your suggestions and measured the thicknesses of the $\text{CuMo}_6\text{S}_8/\text{Cu}$ electrode by vernier caliper and the CuMo_6S_8 layer by cross-section SEM. The thickness of the whole electrode is 0.4 mm and the CuMo_6S_8 layer varies from 1 to 6 μm (Fig. R4). We think the introduction of surface roughness factor may be helpful for fair comparison of performance, but most experiment techniques such as AFM, 3D optical profiler, step profiler, *etc.* are difficult to acquire accurate topography and real surface area of this rugged CuMo_6S_8 layer on porous Cu foam. As is shown in Fig. R1b, the large quantities of nanometer and micrometer-level tortuous pores could make the probe of AFM hard to reach. Moreover, the accuracy of X and Y direction of optical profiler is low with

sub-micrometer level, which may cause large error for measurement of real surface area of CuMo₆S₈ layer. We also notice that in literature, there are rare data about surface roughness of reported catalysts (no matter for relatively flat Pt/C or relatively rough self-supporting catalysts), which makes direct comparison difficult

As you are concerned, the porous structure may bring large quantities of active sites, thus may affect the fair comparison between CuMo₆S₈ and other samples at large current density. Considering the applicability of electric double layer capacitance method for porous materials, we introduce the electrochemical surface area (ECSA) to normalize the performance and realize fair comparison (Fig. R5), which is widely used in the HER field. The results show that the ECSA of CuMo₆S₈/Cu is the highest, and the activity normalized by ECSA is close to that of Pt/C electrode. Besides that, we also compare the mass activity of these three electrodes, and the results show that CuMo₆S₈/Cu electrode could maintain high mass activity of 250 A g⁻¹ at -0.32 V vs RHE, which is nearly equal to that of Pt/C electrode. This is because the CuMo₆S₈/Cu electrode has excellent mass transfer ability at large current density, and could ensure that many active sites been exposed and not masked by hydrogen bubbles. Considering that Pt is the well-HER catalysts with the best intrinsic activity, our results reveal that the CuMo₆S₈ has decent intrinsic activity similar to noble metal Pt. Here we want to emphasize that this work aims for improving large current density performance (not the intrinsic activity) of the catalyst, which is important for the industrial HER applications. Such large current density operation needs to consider many factors, including intrinsic activity of catalyst, robust mechanical stability, unimpeded interfacial electron transfer, good mass transfer ability and large number of active sites, *etc.* The CuMo₆S₈/Cu electrode possesses these features and thus shows decent overall HER performance at large current density.

Changes to the revised SI. Figs. R5a-c were added as Figs. 8d-f in the revised SI.

Figure R4. Thickness measurements of (a) the whole CuMo₆S₈/Cu electrode and (b) cross sectional SEM image of the CuMo₆S₈ layer on Cu foam.

Figure R5. (a) A capacitive current against the scan rate and corresponding C_{dl} values estimated by linear fitting of the plots for the CuMo₆S₈/Cu, MoS₂, and Pt/C electrodes, (b) ECSA normalized LSV curves of the CuMo₆S₈/Cu, MoS₂, and Pt/C electrodes. (c) Mass activity of CuMo₆S₈/Cu, MoS₂, and Pt/C electrodes. These figures were added as Figs. S8d-f in the revised SI.

Comment 3. Overall, the reviewer cannot recommend the manuscript can be published as present. There are quite a few studies on Chevrel phase of CuMo₆S₈ via reaction in between MoS₂ and Cu at high temperature. Novelty of the manuscript is also lack, otherwise state more unique point of their studies. The manuscript should be improved further by major revision or submit more specialized journals.

Response 3. Thank you very much for the comments. Chevrel phase materials have been discovered many years ago and the materials themselves are indeed not new,

which we fully agree with you. As you mentioned, there have been a few reports on the synthesis and theoretical investigations about the Chevrel phase materials. However, in these works, most electrocatalytic performance of Chevrel phase materials are studied in small current density and acid media, but rarely studied in large current density and basic media which is more close to applications. For example, Bae *et al* reported a layered $\text{Cu}_2\text{S}/\text{Cu}_{2.76}\text{Mo}_6\text{S}_8/\text{MoS}_2$ heterojunction deposited on a Cu foil, and constructed the interfaces among the three transition metal sulfides, but not the interface between Chevrel phase and Cu substrate (10.1126/sciadv.1602215). Note that Bae's catalyst can only achieve relatively small current density of 100 mA cm^{-2} and their stability test was done at even smaller current density of 10 mA cm^{-2} . In another work, Strachan *et al* used solvothermal method to synthesize the Chevrel phase nanocrystals for HER with a current density of only 20 mA cm^{-2} (10.1021/acsnm.0c03355).

Our work is more application oriented and there are three unique points. **First**, the falling-off of catalysts, and dominated mass transfer kinetics have accounted for poor performance and stability of most catalysts at large current density. Our work aims at constructing a superior and robust monolithic electrode to solve these issues in hydrogen evolution at industrial-scale current density (not the claim of synthesizing new materials). **Second**, the proposed dual-interfacial engineering provides fresh ideas for designing high performance electrodes in other gas-involving reactions. **Third**, our $\text{CuMo}_6\text{S}_8/\text{Cu}$ monolith catalyst is among the best performance catalysts for HER in terms of current density achieved and stability test current and duration, as shown in detail in the manuscript.

Response to Reviewer #2

Comment. This manuscript presents a novel synthesis approach to generate a binder-free $\text{CuMo}_6\text{S}_8/\text{Cu}$ electrode and evaluates their hydrogen evolution activity under alkaline conditions. The electrodes exhibit high current densities at relatively low overpotentials as well as high mechanic stability, which was evaluated through In-situ total internal reflection imaging. The electrocatalyst-support interfacial binding and electrocatalyst-gas bubble interfacial adhesion force of the electrodes are evaluated using a force analysis model, which is experimentally validated through micro scratch testing and optical microscopy. The authors also include the adsorption free energy calculations for various CuMo_6S_8 facets. It is observed that facets with undercoordinated sulfur atoms have more favorable H adsorption energies. Badger analysis indicates that this is due to a higher electronegativity difference between S and H, leading to stronger S-H bonds. Lastly, density of states and partial density of states calculations of CuMo_6S_8 are used to elucidate the metallic nature of CuMo_6S_8 , which is expected to have a higher electron transfer efficiency than that of semiconducting 2H-MoS_2 . Based on the novelty of the electrode generation, high current densities, and the extensive characterization of electrocatalyst support/ bubble adhesion force, I consider that Nature Communications is an appropriate platform to publish the manuscript. However, the following suggestions will improve the clarity and reproducibility of the work.

Response: Thank you very much for your recommendations. We appreciate the reviewer very much for his/her very carefully reading and by writing that our work “presents the novelty of the electrode generation, high current densities, and the extensive characterization of electrocatalyst support/bubble adhesion force”.

Comment 1. The use of “fast hydrogen evolution” instead of “fast bubble evolution” in the 4th sentence of the abstract seems more appropriate since the reaction under study is currently not mentioned in the abstract at all. The authors should also emphasize that

the electrocatalyst are evaluated under basic conditions in the abstract. This detail increases the significance of the work since Chevrel Phases have been extensively evaluated as HER catalyst in acidic media, but very rarely in basic media.

Response 1. These are very instructive suggestions which we appreciate very much. We gladly followed your two suggestions and revised the abstract as follows.

Changes to the Abstract in the revised manuscript. “Here we construct a mechanically-stable, all-metal, and highly active $\text{CuMo}_6\text{S}_8/\text{Cu}$ electrode by *in-situ* reaction between MoS_2 and Cu. The Chevrel phase electrode is rarely used in alkaline electrolyte previously and exhibits strong binding at electrocatalyst-support interface while weak adhesion at electrocatalyst-bubble interface, in addition to fast hydrogen evolution and charge transfer kinetics.”

Comment 2. The significance of the TEM image shown in Fig. 1h will increase if the reader could identify the (131) and (110) planes in the XRD shown in Fig. 1i. The authors are encouraged to index the index pattern to facilitate this comparison.

Response 2. We have taken your suggestion and indexed the main crystal facets to the pattern as follows.

Figure R6. X-ray diffraction patterns of the samples before and after annealing. This figure has been used as new Fig. 1i in revised manuscript.

Comment 3. To facilitate the comparison between MoS₂ and CuMo₆S₈ of Fig. 1i, the authors are encouraged to provide a zoom-in version of the peak between 10-20 2θ in the SI.

Response 3. Thank you for your suggestions. The zoom-in inspection of the peaks in 2 theta of 10-20 ° was added to Supplementary Fig. S6 as follows.

Figure R7. The zoom-in XRD patterns of the peaks in 2 theta of 10-20 °. This figure has been added as Fig. S6 in revised SI.

Comment 4. References for the overpotentials and current densities included for MoS₂ and Pt/C electrodes in the following sentence should be included: “Fig. 2a shows that the overpotentials of CuMo₆S₈/Cu electrode are only 320 mV at 1000 mA cm⁻² and 334 mV at 2500 mA cm⁻², which are much smaller than those of MoS₂ (579 mV @1000 mA cm⁻²) and Pt/C (474 mV @1000 mA cm⁻²) electrodes.”

Response 4. Thank you for your kind suggestions. Actually, the data of MoS₂ and Pt/C electrodes are tested by ourselves. We have made this point clear in the revised version. We also compared our Pt/C catalyst with other reports and found that it is in the reasonable range as shown below.

Changes to the revised manuscript. Some sentences were added in manuscript as follows. On Page 7, “The performance of Pt/C electrode is also provided as a reference. Here the Pt/C electrode is prepared by dropping Pt/C ink on Cu foam, which shows similar performance compared with literature (Supplementary Table 2).”

Table R1. HER performance of Pt/C electrode at large current density. This table was added as Supplementary Table 2 in revised SI.

Electrocatalysts	η (mV) @ 1000 mA/cm ²	References
Pt/C-NF	490	13
Pt/C-NF	780	14
Pt/C-NF	650 @ 400 mA cm ⁻²	15
Pt/C-NF	370 @ 500 mA cm ⁻²	18
Pt/C-NF	350 @ 460 mA cm ⁻²	20
Pt/C	520 @ 600 mA cm ⁻²	21
Pt/C-NF	420	25
Pt/C	240 @ 400 mA cm ⁻²	28
Pt/C	230 @ 800 mA cm ⁻²	29
Pt/C	670	37
Pt/C	410	1
Pt/C-Cu foam	450	This work

Comment 5. The authors should include the specific capacitance used to calculate the ECSA of the electrodes evaluated as well as the equation used to convert the applied potential from Hg/HgO to RHE.

Response 5. Thank you for your suggestions. The specific capacitance and the equation used to convert potential from Hg/HgO to RHE have been added to the experimental section in SI.

Changes to the revised SI. We added the following sentences on Page 3 of SI.

- 1) “The specific capacitances of CuMo₆S₈ and MoS₂ are 0.04 mF cm⁻², and that of Pt is 0.196 mF cm⁻².”
- 2) “The following equation was used to convert the applied potential from vs Hg/HgO to vs RHE: $E_{vs\ RHE} = E_{vs\ Hg/HgO} + 0.059 * pH + 0.098$.”

Comment 6. Since the overpotential at 10 mA/cm² has been extensively used as a figure of merit for HER catalyst, the authors should include it in the manuscript or SI to facilitate the comparison with other HER catalysts besides the ones included in table 2 of the SI.

Response 6. Thank you for your suggestions. The LSV curves of three electrodes in the range of small current density are shown in Fig. R8.

Changes to the revised manuscript. Fig. R8 was added as Fig. 8c in the revised SI, and the overpotential at 10 mA cm⁻² was included in the manuscript as follows. On Page 7, “The overpotential of CuMo₆S₈/Cu electrode at 10 mA cm⁻² is 172 mV, and the intrinsic activity of CuMo₆S₈ is obtained by normalizing electrochemical surface area (ECSA) and mass of electrocatalyst (Supplementary Figs. 8c-f).”

Figure R8. The LSV curves of three electrodes at small current density.

Comment 7. The authors should include the potential required to achieve the current densities in figure 2c.

Response 7. Thank you for your suggestions. The potential required to achieve the current densities have been added to the caption of Figure 2 in the revised manuscript as follows.

Changes to the revised manuscript. On Page 9, “Chronoamperometric (i-t) curves of the CuMo₆S₈/Cu electrode at current densities of -500, -1000, -2500 mA cm⁻² over

300 h, where the current densities correspond to potentials of -0.43 V, -0.57 V, -0.94 V vs RHE without iR correction, respectively.”

Comment 8. The H-S bond strength argument obtained from the badger charge analysis has been previously elucidated in Chevrel Phase chalcogenides and should be appropriately cited (<https://doi.org/10.1021/acsami.0c07207>).

Response 8. Thank you for pointing out this nice theoretical paper. This literature has been cited as Ref. 43 in the revised manuscript and discussed as follows.

On Page 15, “The effect of chalcogen electronegativity of Chevrel phase chalcogenides (Mo_6X_8 ; X= S, Se, Te) on HER activity has been investigated previously⁴³. Here we further study the relationship between the coordination of sulfur atoms on main crystal facets of CuMo_6S_8 and its HER activity.”

Response to Reviewer #3

Comment. In this manuscript, a mechanically stable Chevral phase $\text{CuMo}_6\text{S}_8/\text{Cu}$ electrode for electrocatalytic hydrogen evolution reaction is reported. The authors claimed that the dual interfacial engineering strategy enables resistance against peeling-off of electrocatalysts and improved reaction kinetics at large current density, resulting in better HER performance. The results are potentially interesting to readers in the field. However, there are still some technical issues that require clarification.

Response : Thank you very much for your recommendations. We appreciate the reviewer by writing that our work reports “better HER performance, potentially interesting to readers in the field.”

Comment 1. In-situ total internal reflection imaging method was used to study the stability difference between Pt/C and $\text{CuMo}_6\text{S}_8/\text{Cu}$ electrodes before and after 10,000 cyclic voltammetric scans. Due to the evanescent field imaging requirement, the electrode surface must be in close contact with the prism. Considering that HER is a three-phase reaction, the diffusion of the hydrogen gas and the generation of bubbles have pivotal role in the process. In the experiment, the prism has great influence on the diffusion of hydrogen gas. The weak bubble adhesion at the electrode surface seems to be one of the key selling points of the research. However, bubble adhesion on the prism was not well discussed, which might be the major contributing cause in observed results. These two factors, blocked hydrogen diffusion and surface bubbles on the prism, could lead to apparent deviation of testing condition from the real application.

Response 1. Thank you very much for these instructive suggestions. We totally agree with the reviewer that the prism would affect hydrogen diffusion and bubbles adhesion on its surface. However, according to our test principle and repeated experiment verification, these two issues would not affect our final results of onset potential. **First**, our test principle is mainly based on total internal reflection (TIR). When the incident light radiates the prim-electrolyte interface at the critical angle, TIR will occur and

generate an evanescent wave with a certain penetration depth (~ 637 nm). When the potential applied to the electrode reaches the onset potential, H₂ microbubbles will be generated and lead to a change in the equivalent refractive index (RI), resulting in a sudden change in the reflected light intensity. It is worth noting that our onset potential results are only obtained in this early instantaneous process, where these microbubbles are invisible due to their small sizes and very small amounts. This is like a “turn on” moment of the HER reaction. Note that the growth of microbubbles into big bubbles, as well as the subsequent diffusion and adhesion to the prism all occur after the process of obtaining data, so it would not affect the test results. According to our previous work, the RI resolution of our method could reach 8.13×10^{-7} refractive index unit (RIU) (10.1109/TIM.2019.2928349, 10.1016/j.snb.2018.12.023), which means the onset potential can be obtained when the volume concentration of H₂ bubbles reaches a very small number of 1.988×10^{-6} in evanescent wave layer (Table R2, see more details in Note R1). To verify the above analysis, we conducted more experiments and a test video is provided in Supplementary Video. We find that the light intensity corresponding to point A and B in the Pt/C electrode has changed abruptly at -0.072 and -0.055 V vs RHE at the “turn on” moment of the reaction. At this time, there was no visible bubble diffusion and adhesion to prism, indicating that they had no effect on our test results.

Second, inspired by your comment #2, we performed multiple experiments on different samples and they show consistent conclusion (see details below). These parallel experiments also support the validity of our test method.

Changes to the revised manuscript. Some sentences have been added in revised manuscript for further description as follows.

1) On Page 10, “It is worth noting that in this method, the onset potential results are obtained in the early instantaneous process, where the microbubbles are invisible due to their small sizes and very small amounts. This is like a “turn on” moment of the HER reaction. Note that the growth of microbubbles into big bubbles, as well as the subsequent diffusion and adhesion of bubbles to the prism all occur after the process of obtaining onset potential data, so it would not affect the test results (Supplementary Video).”

2) On Page 11, “In addition, the results above and the repeatability of the experiments are also verified by three parallel experiments for different CuMo₆S₈/Cu and Pt/C electrodes (Supplementary Figs. 14-16), as well as three replicate experiments on the same CuMo₆S₈/Cu electrode (Supplementary Fig. 17).”

3) On Page 11, “In addition, the TIR method also has unique advantages of large field of view, high detection sensitivity for hydrogen bubble formation (Supplementary Table 6, Note 3), easy operation and low requirements for equipment, compared with other optical imaging methods such as X-ray imaging.”

Note R1. The relationship between equivalent RI/density and volume concentration of H₂ bubbles follow the equations below (10.1002/sml.202102407):

$$n_{\text{equivalent}} = (1 - v) \times n_{\text{electrolyte}} + v \times n_{\text{bubble}} \quad (1)$$

$$\rho_{\text{equivalent}} = (1 - v) \times \rho_{\text{electrolyte}} + v \times \rho_{\text{bubble}} \quad (2)$$

Where n is the RI, ρ is the density and the v is the volume concentration of H₂ bubbles. Amongst, the RI and density of 1M KOH and H₂ are known, and the RI resolution and density resolution of TIR and X-ray imaging methods are obtained from literature, respectively. Therefore, the equivalent RI and density can be obtained by subtracting the RI resolution and density resolution from the RI and density of KOH. Finally, the volume concentration of H₂ bubbles could be obtained according to the equation (1) and (2). The results are shown in Table R2 below. The Note R1 was added as Supplementary Note 3 in revised SI.

Comment 2. The imaging technique could still provide some insight of the structure and activity change of the electrode surface. Unlike the previous carbon fiber experiment, the structures of which can be well resolved, the surface of the electrodes are almost homogenous and very random. Information from a single or a few in-situ experiments is not very convincing. Multiple parallel experiments and statistical analysis of the results would be sufficient.

Response 2. Thank you very much for your nice suggestions. We followed your

suggestions and carried out additional three parallel experiments and statistical analysis for CuMo₆S₈/Cu and Pt/C electrodes (Figs. R9-R11). The results from the three parallel experiments are quite similar. As is shown in statistical analysis of Fig. R11, the frequency represents the proportion of the certain potential difference pixels to the total image area. The area of large difference values > 0.1 V occupies about 92% field of the Pt/C electrode. Inversely, the area of small difference values < 0.1 V accounts for about 85% for the CuMo₆S₈/Cu electrode, indicating that the CuMo₆S₈/Cu electrode has much better stability than the Pt/C electrode. Besides that, we also carried out three replicate TIR experiments on the same CuMo₆S₈/Cu electrode (Fig. R12). The three onset potential mappings are nearly identical, suggesting the high reproducibility of this technique.

Changes to the revised SI. Figs. R9-12 were added as Figs. S14-17 in revised SI.

Figure R9. The results of three parallel experiments of Pt/C electrodes. Figs. a1, b1 and c1 show the onset potential mapping of Pt/C electrodes before 10,000 CVs, while figs. a2, b2, and c2 show that after 10,000 CVs. Figs. a3, b3, and c3 show the absolute

potential differences between the two before and after 10,000 CVs tests. This figure was added as Fig. S14 in revised SI.

Figure R10. The results of three parallel experiments of $\text{CuMo}_6\text{S}_8/\text{Cu}$ electrodes. Figs. a1, b1 and c1 show the onset potential mapping of three $\text{CuMo}_6\text{S}_8/\text{Cu}$ electrodes before 10,000 CVs, while Figs. a2, b2 and c2 show that after 10,000 CVs. Figs. a3, b3 and c3 show the absolute potential differences between the two before and after 10,000 CVs tests. This figure was added as Fig. S15 in revised SI.

Figure R11. Statistics of absolute differences of onset potential from three parallel experiments on $\text{CuMo}_6\text{S}_8/\text{Cu}$ (a) and Pt/C (b) electrodes, and their comparison (c). This figure was added as Fig. S16 in revised SI.

Figure R12. Onset potential mapping results of three replicate measurements on the same $\text{CuMo}_6\text{S}_8/\text{Cu}$ electrode. This figure was added as Fig. S17 in revised SI.

Comment 3. The spatial resolution of the imaging technique is at micrometer scale, which is in the same range of other well-developed in situ techniques. Operando X-ray imaging, which offers similar resolution and minimum interference to the testing cell, could be a better alternative to the current method.

Response 3. Thank you for this instructive comment. We agree that operando X-ray imaging is a powerful tool as you mentioned. We think there are some unique points of our method. **First**, compared with centimeter level field of view of TIR, the view of sub-micrometer level in typical X-ray imaging is not suitable for evaluating large area self-supporting electrode. Actually, we believe that X-ray imaging is more suitable to study of the single gas bubble (10.1016/j.elecom.2015.03.009, 10.1038/s41467-021-23664-1), while our method is more suitable to evaluate the performance of whole electrodes. **Second**, the focus of our method is not only on imaging, but also on measuring the onset potential of HER. Nowadays, the density resolution of X-ray imaging is in the

range of 0.3-10 mg cm⁻³ (10.1038/srep05332, 10.1088/1742-6596/425/19/192007). Taking the density resolution of 0.3 mg cm⁻³ as an example, the detectable H₂ volume concentration is 2.86 × 10⁻¹ (Table R2, see more details in Note R1), which means the H₂ bubble detection sensitivity of X-ray imaging is two orders of magnitude lower than that of TIR (1.99 × 10⁻⁶). Therefore, it is more accurate for our method to obtain onset potential of electrode according to hydrogen bubble formation. **Third**, compared with the complex equipment required for X-ray imaging, such as synchrotron radiation source and radioactivity, the TIR method is simple, easy to operate, and not radiative, so it is more applicable for routine electrochemical reactions. In short, we believe that the TIR method has unique points in the application of self-supporting electrodes and is expected to expand to other electrochemical fields.

Table R2. Volume concentration of H₂ bubble calculated from RI and density resolution. This table has been added as Table 6 in revised SI.

	1M KOH	H ₂	Resolution	Equivalent RI/ Density	Volume concentration
RI	1.409	1.0001	8.13E-07*	1.4089	1.99E-06
Density (g cm ⁻³)	1.05	8.90E-05	3.0E-04#	1.047	2.86E-04

* From DOI: 10.1109/TIM.2019.2928349

From DOI: 10.1088/1742-6596/425/19/192007

Response to Reviewer #4

Comment. In the manuscript, the author designed a stable and highly active CuMo₆S₈/Cu electrode by in-situ reaction between MoS₂ and Cu. The strong binding at electrocatalyst-support interface while weak adhesion at electrocatalyst-bubble interface made CuMo₆S₈ reach large current density and highly stable in Hydrogen Evolution Reaction (HER). However, the catalysts, the device performance and the claimed novel investigation method have been reported. Furthermore, the mechanism and intrinsic activity is not clear and in-depth in this research. Considering the novelty of this manuscript, I recommend reconsider after major revision, some points for the author.

Response: Thank you very much for your recommendations. We appreciate the reviewer by commenting that our catalyst “is stable and highly active” and “reach large current density and highly stable”.

Comment 1. In supplementary Fig. 2, the morphology of CuMo₆S₈ is not uniform, and there is also a ribbon-like structure, which need to be confirmed.

Response 1. Thank you very much for your careful reading. We have followed your question and carefully characterized the morphology again and confirmed that the CuMo₆S₈ electrode is a porous structure formed by stacking of nanosheets or nanoparticles, and mixed with a few nanowires (ribbon-like structure), as shown in Fig. R13. We also analyzed the nanowires by EDS mapping per your request, and the results showed that there are elements of Cu, Mo, and S in the nanowire (Fig. R14). Besides that, we also checked the X-ray diffraction pattern, and determined that there is no other impure phase in CuMo₆S₈ catalysts. Therefore, we ruled out the possibility that the nanowires were impurities and determined that they were also CuMo₆S₈. Here we speculate that the morphological diversity of the material may be due to the low melting point of Cu and its easy diffusion during high temperature reaction, leading to the growth of nanowire materials.

Changes to revised manuscript and SI. Some sentences were added in manuscript as follows.

On Page 4, “After annealing, scanning electron microscopy (SEM) images show that the morphology of materials changes from agglomerated nanoflakes to porous structure, which composed of nanoflakes or nanoparticles and a few nanowires. (Supplementary Figs. 2d, 4). The morphological diversity of the material may be due to the low melting point and high temperature diffusion of Cu.” In addition, Figs. R13a, b and R14 were added as Figs. S4a-g in revised SI.

Figure R13. SEM images of the $\text{CuMo}_6\text{S}_8/\text{Cu}$ electrode. The porous structure is formed by stacking of nanosheets or nanoparticles (a-b), and mixed with nanowires (c-d). Fig. R14a, b were added as Figs. S4a, b in revised SI.

Figure R14. (a) SEM image of the $\text{CuMo}_6\text{S}_8/\text{Cu}$ nanowire. (b-e) EDS element mappings of the $\text{CuMo}_6\text{S}_8/\text{Cu}$ nanowire. These figures were added as Figs. S4c-g in revised SI.

Comment 2. The particle size of Pt/C is not given in the manuscript, and the particle size of CuMo_6S_8 and MoS_2 is different, which may also affect the performance comparison.

Response 2. Thank you for your suggestions. We have taken your suggestion and conducted SEM characterization to measure the particle sizes of Pt/C (Fig. R15). The average size of Pt/C is about 150 nm, and small nanoparticles tend to agglomerate to large ones due to the link of Nafion binder. We find that the morphology of CuMo_6S_8 and MoS_2 is different and they are not particle shape, making it difficult to compare the “particle size” among three catalysts. We think it is more meaningful to compare the performance of catalyst. To compare their performance, in addition to the electrode area normalized performance in original manuscript, here we also compared the performance normalized by electrochemical surface area (ECSA, Fig. R16). The results show that the ECSA of CuMo_6S_8 is improved largely compared with MoS_2 in case of the same loading, but the intrinsic activity of CuMo_6S_8 is still much better than MoS_2 . Besides, the intrinsic activity of CuMo_6S_8 is also close to Pt/C. Considering that Pt is recognized as one of catalysts with the best intrinsic activity, these results reveal that the CuMo_6S_8 has decent intrinsic activity. In addition, we also want to emphasize that this work aims for the industrial HER applications, which requires not only good intrinsic activity of materials, but also good overall performance of electrodes,

including robust mechanical stability, unimpeded interfacial electron transfer, good mass transfer ability and large number of active sites, *etc.*

Changes to the revised SI. Fig. R16 was added as Figs. S8e, f in revised SI.

Figure R15. (a-c) SEM images of Pt/C catalysts at different magnifications. (d) Particle size distribution of Pt/C catalysts.

Figure R16. (a) A capacitive current against the scan rate and corresponding C_{dl} values estimated by linear fitting of the plots for the CuMo₆S₈/Cu, MoS₂, and Pt/C electrodes. (b) ECSA normalized LSV curves of the CuMo₆S₈/Cu, MoS₂, and Pt/C electrodes. These figures were added as Figs. S8e, f in revised SI.

Comment 3. The XPS results showed the presence of S-Mo and S-Cu bonds. Is it possible that Mo-O bonds or S-O bonds are formed at the interface due to the presence of Cu₂O according to the XRD results?

Response 3. This is a very interesting point. We think it is possible that Mo-O bonds or S-O bonds are formed at the interface. In terms of experimental characterization, we find that it is difficult to verify this point. For XPS, **first**, the spatial resolution of XPS is limited to spot size of X-ray (~100 μm in our facility). Considering that both CuMo₆S₈ and Cu foam itself are rough over 100 μm size level, it is difficult for XPS covering 100 μm size level to get the information only from the interface while not the near surface information of the electrode. **Second**, we also tried to get XPS depth signals of interface by ion etching. However, the rough and porous CuMo₆S₈ layer cannot be uniformly thinned, which means that besides the XPS signals from the interface, signals from the surface CuMo₆S₈ residue and internal Cu substrate may be also included. Therefore, based on the discussion above, we think it may be very difficult to verify this point by XPS.

We also tried hard to get more microscopic information of the interface between CuMo₆S₈ and Cu by HRTEM (Fig. R17). The dashed line represents the crystal boundary between CuMo₆S₈ and Cu. Occasionally, we observe Cu₂O phase as shown by the circled area. This result shows that there are extremely small Cu₂O phases existed in the interface between CuMo₆S₈ and Cu, presumably due to the easy oxidization of copper. However, HRTEM imaging cannot tell whether there is Mo-O or S-O bond at interface.

Figure R17. HRTEM image of the $\text{CuMo}_6\text{S}_8/\text{Cu}$ electrode. The dashed line represents the boundary between CuMo_6S_8 and Cu , and the circled area stands for the Cu_2O phase.

Comment 4. The loading mass of the catalyst is an important factor affecting the HER activity. Whether the loading mass of $\text{CuMo}_6\text{S}_8/\text{Cu}$ and Pt/C is the same, it is recommended to supplement the mass activity of the catalysts.

Response 4. Thank you for your suggestions. The loading mass of $\text{CuMo}_6\text{S}_8/\text{Cu}$ is 10 mg cm^{-2} . We have followed your suggestion and performed more experiments with different loading mass of Pt/C (0.5 mg cm^{-2} as shown in our original manuscript and in most literature, and also new experiments with 10 mg cm^{-2} in Fig. R18a). The results show that the Pt/C electrode with the loading of 10 mg cm^{-2} shows better performance than CuMo_6S_8 below 1750 mA cm^{-2} , while worse above 1750 mA cm^{-2} . We also find that the Pt/C electrodes have relatively poorer mass transfer ability at large current density than CuMo_6S_8 , as indicated by $\Delta\eta/\Delta\log|j|$ ratios at large current density (Figs. R18b-c). However, the major problem for Pt/C catalyst with high loading of 10 mg cm^{-2} is its poor stability (Figs. R18d-f). Compared with the Pt/C with 0.5 mg cm^{-2} , that with 10 mg cm^{-2} exhibits nearly 4 times of degradation rate ($8 \mu\text{A s}^{-1}$). Actually, in literature, Pt/C with high loading of 10 mg cm^{-2} is rarely used due to its poor stability and high cost.

To compare the performance of CuMo_6S_8 and Pt/C fairly, we followed your

suggestion and used the mass activity to normalize the performance (Fig. R19). The results show that CuMo₆S₈/Cu electrode maintains high mass activity of 250 A g⁻¹ at -0.32 V vs RHE, which is nearly equal to that of Pt/C electrode. This is because the CuMo₆S₈/Cu electrode has excellent mass transfer ability at large current density, and could ensure that many active sites are exposed and not masked by hydrogen bubbles. Considering that Pt is the well-HER catalysts with the best intrinsic activity, our results reveal that the CuMo₆S₈ has decent intrinsic activity similar to noble metal Pt.

Changes to the revised SI. Figure R19 was also added as Fig. S8d in revised SI.

Figure R18. (a) LSV curves of CuMo₆S₈/Cu, MoS₂, Pt/C electrodes with the loading of 10 mg cm⁻². Δη/Δlog|j| ratios of CuMo₆S₈/Cu, MoS₂ and Pt/C electrodes with 0.5 mg cm⁻² (b) and 10 mg cm⁻² (c) at different current density ranges. (d) CA curves of Pt/C electrodes with the loading of 10 mg cm⁻² and 0.5 mg cm⁻². LSV curves of Pt/C electrodes with the loading of 10 mg cm⁻² (e) and 0.5 mg cm⁻² (f) before and after CA test.

Figure R19. Mass activity of CuMo₆S₈/Cu, MoS₂, and Pt/C electrodes. This figure was added as Fig. S8d in revised SI.

Comment 5. The author the HER active sites on three main facets of CuMo₆S₈, showing the CuMo₆S₈ (001), (101) and (110) facets are excellent active sites for HER, which is the best active site?

Response 5: Thank you for your questions. The ideal HER electrocatalyst should show a moderate adsorption free energy of H* (ΔG_{H^*}) close to 0 eV. According to our DFT calculations of ΔG_{H^*} , S atom with coordinate number of 3 ($\Delta G_{H^*} = 0.084$ eV) on CuMo₆S₈ (001) is the best active site among all the considered sites.

Comment 6. The author studied the effect of facets and coordinate number of S atom to performance, which is the most essential factor leading to the good performance of the CuMo₆S₈, what's the relationship between them?

Response 6: Thank you for your questions. Both facet and coordinate number of S are essential factors leading to good performance. It is difficult to conclude which one is more important because they are associated with each other. Here we emphasize that the coordinate number of S is the descriptor of HER activity for each facet considered in this study. By combining the free energy diagrams of Figure 5e, Supplementary Figure 21 and the Bader effective charge of Supplementary Table 6, we conclude that the S atom with larger coordinate number shows more negative Bader effective charge

and higher H^* adsorption free energy. Since the S atom is more electronegative than H in S-H bond, a larger coordinate number of S (already has more negative Bader effective charge) should show a weaker S-H bond strength, *i.e.*, a higher ΔG_{H^*} . It is also noted that this rule only works for S atoms on the same facet but not on different facets.

Changes to the revised manuscript: Some sentences were added in the revised manuscript as follows. On Page 15, “From this perspective, the coordinate number of S atom is a descriptor of HER activity for each facet considered in this study. Noted that this rule only works for S atoms on the same facet but not on different facets.”

REVIEWER COMMENTS

Reviewer #1 (Remarks to the Author):

The authors replied the reviewer's comment 1 & 2 in partly satisfaction. Please refer to JACS 2015, 137,4347. Please make results of electrocatalytic performances, in specially, the stability tests. It is not guaranteed for the stable catalyst of around 100 hours operation even at the large current density, since the larger electrochemical surface area provides resulting large current density. At the same surface area and $j = 10 \text{ mA/cm}^2$ as followed the benchmarking protocol.

The most critical concern is that the novelty issue. Complete literature survey is indeed required to support their novelty. Please include the survey with criticism in introduction. As replied in their response "As you mentioned, there have been a few reports on the synthesis and theoretical investigations about the Chevrel phase materials. However, in these works, most electrocatalytic performance of Chevrel phase materials are studied in small current density and acid media, but rarely studied in large current density and basic media which is more close to applications." it is not acceptable. "large current density" and "basic media" never warrant the manuscript can be published. Please refer and summarize what they referred to in their response. Bae et al. and Strachan et. al. should be included in their literature survey to make their work novel. The reviewer cannot agree with "more application oriented" and understand "there are three unique points." Three unique points are neither unique nor scientific. 1. current density should be normalized by electrochemical surface area and 2. stability test should be more than 1 week under the operation. Refer to Bae et al. almost 1 month stability test done.

The manuscript still failed to show the soundness of the research they've reported. It cannot be published in Nature Comm. as it is.

Reviewer #2 (Remarks to the Author):

The authors have exhaustively addressed most of the comments from the first round of revisions. However, it is still unclear if the specific capacitance values used to calculate ECSA have been obtained from literature or have been calculated as part of the manuscript. The authors should either provide the appropriate references or explain the method used to calculate the specific capacitance of CuMo_6S_8 , MoS_2 , and Pt. Especially since CuMo_6S_8 and MoS_2 share the same specific capacitance value.

Reviewer #3 (Remarks to the Author):

This manuscript can be recommended for publication without further revision.

Reviewer #5 (Remarks to the Author):

Overall, it appears that the authors managed to implement very good the concept of “structure engineering over catalytic activity”. In other words, they prepared an “average” catalyst from the intrinsic point of view (the data at low overpotentials are comparable to what is reported in the literature for CP and defected MoS₂) but with the remarkable property of an extremely aerophobic surface that decreases dramatically the ohmic drop at high overpotentials. The latter allows the material to maintain its initial catalytic activity and in fact at very high current densities both Volmer and Heyrovsky steps are catalyzed giving rise to a decrease in the Tafel slope from ca. 120 to less than 30 mV/dec. This is in contrast to the general consideration for noble metal catalysts such as Pt, where an increase in overpotential (or applied current density) results in an increase in Tafel slope. The latter finding is very interesting because it can potentially provide novel mechanistic insights into HER over non-noble metal catalysts.

I have a few rather minor comments mainly about the nature of the material that I append right below (also, most of my initial comments were answered by the existing thorough review comments).

1. In the preparation procedure, it is not clear how the authors adjusted the stoichiometry of the material since, the amount of Cu reacted is practically unknown (the reaction was carried out on bulk Cu with just MoS₂).
2. In the TEM data it would be useful to provide the lattice spacing for the (101) and (111) planes and compare the results with what is reported in the literature for the CP (I personally haven't seen any data for the (131) plane). Furthermore, MoS₂ has an interplanar distance of ca. 0.27 nm. Can the authors prove that the interplanar distance ascribed to the (131) plane is not related to the presence of modified MoS₂ (e.g., Cu-doped)?
3. In the XRD data there is a shift of the MoS₂ peak from 14.4° to 13.7° that the authors attribute to the conversion of MoS₂ to CP. In line with the above comment (#2), the unclear features in the XRD diffractogram (possibly related to the relatively thin thickness of the film) and considering the experimental procedure followed, how can the possibility of Cu-doped MoS₂ formation (instead of CP) be excluded?
4. In the XPS spectrum of Mo 3d (5/2), there is a shift to lower binding energy values (ca. 2 eV) compared to what is reported in the literature for non-leached CP (for example in our paper). This might imply a charge transfer from the “doping/impurity” element (i.e., Cu) towards Mo. How this finding is compared to a Cu-doped MoS₂ sample?
5. The method used for the determination of ECSA is not clear. I understand that the 40 μF/cm² specific capacitance value for CP and MoS₂ is based on the theoretical capacitance for monolayer MoS₂ (which renders the approach highly approximate) but where does the value of 196 μC cm⁻² for Pt come from?

Response to Reviewer #1

Comment 1. The authors replied the reviewer's comment 1 & 2 in partly satisfaction. Please refer to JACS 2015, 137, 4347. Please make results of electrocatalytic performances, in specially, the stability tests. It is not guaranteed for the stable catalyst of around 100 hours operation even at the large current density, since the larger electrochemical surface area provides resulting large current density. At the same surface area and $j = 10 \text{ mA/cm}^2$ as followed the benchmarking protocol.

Response 1. Thank you very much for your comments. The method in this literature (JACS 2015, 137, 4347) is a classic protocol for testing the change of intrinsic activity of catalyst. Therefore, we have followed your suggestions and used the ECSA normalized LSV curves to reflect the change of intrinsic activity of our catalyst before and after CA test of 300 h (Fig. R1, Table R1). As shown in Table R1, the change of overpotentials at ECSA normalized current densities before and after the CA test does not exceed 0.9 %. The degree of change is of the same order of magnitude as reported in the literature (JACS 2015, 137, 4347).

In addition, we agree that the larger electrochemical active area can improve the geometrical current density. Note this cannot necessarily lead to a stable catalyst at large geometrical current density. As shown by Fig. R2 and our recent work (Fundamental research, 2022, doi.org/10.1016/j.fmre.2022.03.017), the higher loading of Pt/C electrode leads to improved geometrical current density, but the falling-off of catalyst is more serious, resulting in a poorer stability. This also shows that solely improving active sites cannot ensure the stability of electrode at large current density, and effective structure engineering should be considered. As commented by #5 reviewer, “very good in concept of ‘structure engineering over catalytic activity’”.

Besides, we would like to clarify the difference between the stability test under ECSA normalized current density and large geometric current density. The change of electrochemical activity of the catalysts can be truly investigated when tested under the ECSA normalized current density. While the impacts for the catalyst stability of harsh

electrochemical environment can be reflected when tested at large geometric current density. For example, assuming two identical catalysts with ECSA of 1 cm^2 and 1000 cm^2 , in order to ensure the same ECSA normalized current density of 10 mA cm^{-2} , the tested current values for them are 10 mA and 10 A . In this case, the former can be only constrained by Tafel kinetic, with little affected by mass transfer limited process; while the latter will face tougher electrochemical environment, including strong bubble evolution, convection flow, polarization electric field, and thermal effect. Therefore, the effectiveness of structural engineering can be better investigated by using large geometrical current density. Besides, for our sample with large ECSA of 14392 cm^2 , the equipment applied current will go up to 143 A in order to reach a ECSA normalized current of 10 mA cm^{-2} , which is practically not possible to apply in laboratory. Actually in literature, researchers rarely use such ECSA normalized current density to test stability for large current density electrodes.

Changes to revised SI. Fig. R1 and Table R1 have been added on Page 17 and 23 in the revised SI.

Figure R1. ECSA normalized LSV curves of the CuMo_6S_8 electrode before and after CA test of 300 h. This figure has been added on Page 17 in the revised SI.

Table R1. Parameters for ECSA normalized LSV curves of the CuMo₆S₈ electrode before and after i-t test. This table has been added on Page 23 in the revised SI.

	$\eta@-0.0005$ mA cm ⁻² (ECSA)	$\eta@-0.001$ mA cm ⁻² (ECSA)	$\eta@-0.01$ mA cm ⁻² (ECSA)	$\eta@-0.1$ mA cm ⁻² (ECSA)
Before i-t	-0.1842	-0.2962	-0.2897	-0.3166
After i-t	-0.1843	-0.2971	-0.29	-0.3195
$\Delta(\%)$	0.05%	0.30%	0.10%	0.91%

Figure R2. (a) CA curves of Pt/C electrodes with the loading of 10 mg cm⁻² and 0.5 mg cm⁻². LSV curves of Pt/C electrodes with loading of 10 mg cm⁻² (b) and 0.5 mg cm⁻² (c) before and after CA test for 7200s. Higher loading will lead to a better performance but also poor stability.

Comment 2. The most critical concern is that the novelty issue. Complete literature survey is indeed required to support their novelty. Please include the survey with criticism in introduction. As replied in their response “As you mentioned, there have been a few reports on the synthesis and theoretical investigations about the Chevrel phase materials. However, in these works, most electrocatalytic performance of Chevrel phase materials are studied in small current density and acid media, but rarely studied in large current density and basic media which is more close to applications.” it is not acceptable. “large current density” and “basic media” never warrant the manuscript can be published. Please refer and summarize what they referred to in their response. Bae et al. and Strachan et. al. should be included in their literature survey to make their work novel. The reviewer cannot agree with “more application oriented” and understand

“there are three unique points”. Three unique points are neither unique nor scientific. 1. current density should be normalized by electrochemical surface area and 2. stability test should be more than 1 week under the operation. Refer to Bae et al. almost 1 month stability test done. The manuscript still failed to show the soundness of the research they've reported. It cannot be published in Nature Comm. as it is.

Response 2. Thank you very much for your suggestions. First, we have followed your comments, cited these references, and included the survey with criticism in the Introduction part as follows. “In previous work, nano-Chevrel phase³³ and layered Cu₂S/Cu_{2.76}Mo₆S₈/MoS₂ heterojunction³⁴ were synthesized to study their HER activity. These catalysts delivered geometrical current density of 20 and 100 mA cm⁻², which are relatively small. By optimizing the electrode structure, high performance and good stability of Chevrel phase catalysts at large current density could be realized.”

Second, we want to reaffirm the significance of our work. (i), this work focuses more on the study of the catalyst performance at large current density, not just the intrinsic activity of the catalysts. To realize the high performance and stability of catalysts at large current density is a key to moving towards industrial applications (Adv. Mater., 2022, 34, 2108133). The scientific problems in water splitting at large current density involves mass transfer (Joule, 2020, 4, 555-579; Joule, 2020, 4, 555-579), surface chemistry (Nat. Commun., 2019, 10, 269), interfacial electron transfer (Nat. Commun., 2021, 12, 6051), and stability issues (Adv. Funct. Mater., 2022, 32, 2201726), etc. In this work, the Tafel slope was expanded to a wider range of current density to reflect mass transfer issue. Superhydrophilic and all-metallic design show optimized surface chemistry and interfacial charge kinetics. The viewpoint of strengthening interfacial binding between catalyst and substrate and weakening interfacial tension between catalyst and bubble benefits the mechanical stability of the electrode. The above methods are all effective in solving scientific problems. (ii), to solve the problems of mechanical stability of electrode under large current density from a structural engineering perspective remains a grand challenge. The proposed dual interfacial engineering in this work can provide a new and effective strategy to construct stable and high-performance electrodes for HER and other gas-involving reactions. We note

that other reviewers also support the novelty and significance of our work. For example, as commented by the #3 and #5 reviewers, “The results are potentially interesting to readers in the field”, “implement very good the concept of ‘structure engineering over catalytic activity’”, “the latter finding is very interesting because it can potentially provide novel mechanistic insights into HER over non-noble metal catalysts”. (iii), we introduced an in-situ internal total reflection technology to visualize electrode stability from the perspectives of dynamic imaging and quantitative analysis, which can be complementary to electrochemical methods and benefit the community.

Third, as we mentioned above, it is better to evaluate the catalyst stability under harsh electrochemical environment by using large geometric current density. To our best knowledge, the stability test under 2.5 A in this work is among the highest reported in literature. Moreover, our stability test has been performed for 300 h (nearly two weeks), which is also pretty long considering it was done at such large current density (tougher electrochemical environment). We hope that the reviewer will be satisfied by our detailed explanations above.

Changes to the Introduction of revised manuscript. “In previous work, nano-Chevre phase³³ and layered Cu₂S/Cu_{2.76}Mo₆S₈/MoS₂ heterojunction³⁴ were synthesized to study their HER activity. These catalysts delivered geometrical current density of 20 and 100 mA cm⁻², which are relatively small. By optimizing the electrode structure, high performance and good stability of Chevrel phase catalysts at large current density could be realized.”

Response to Reviewer #2

Comment. The authors have exhaustively addressed most of the comments from the first round of revisions. However, it is still unclear if the specific capacitance values used to calculate ECSA have been obtained from literature or have been calculated as part of the manuscript. The authors should either provide the appropriate references or explain the method used to calculate the specific capacitance of CuMo₆S₈, MoS₂, and

Pt. Especially since CuMo_6S_8 and MoS_2 share the same specific capacitance value.

Response. Thank you very much for your recommendation and nice suggestions. For MoS_2 , the specific capacitance value of $40 \mu\text{F cm}^{-2}$ is used, according to the theoretical capacitance of monolayer MoS_2 ($30\text{-}60 \mu\text{F cm}^{-2}$), and the references (e.g., J. Mater. Chem. A, 2016, 4, 6824; Mater. Today Chem., 2019, 14, 100207; Adv. Funct. Mater. 2018, 28, 1807086). For Chevrel phase CuMo_6S_8 , the same specific capacitance as MoS_2 is used according to the reference (ACS Appl. Mater. Interfaces 2020, 12, 32, 35995), since they have similar chemical components and active sites. For Pt, the specific capacitance value of $196 \mu\text{F cm}^{-2}$ is used according to the references (e.g., Electrochimica Acta, 2005, 50, 2469; Electrochimica Acta, 2017, 224, 468; Int. J. Electrochem. Sci., 2011, 6, 4454).

Changes to revised SI. The above references were added on Page 26 in the revised SI to make these points clear.

Response to Reviewer #3

Comment. This manuscript can be recommended for publication without further revision.

Response. Thank you very much for your recommendation.

Response to Reviewer #5

Comment. Overall, it appears that the authors managed to implement very good the concept of “structure engineering over catalytic activity”. In other words, they prepared an “average” catalyst from the intrinsic point of view (the data at low overpotentials are comparable to what is reported in the literature for CP and defected MoS_2) but with the remarkable property of an extremely aerophobic surface that decreases dramatically the ohmic drop at high overpotentials. The latter allows the material to maintain its initial catalytic activity and in fact at very high current densities both Volmer and Heyrovsky

steps are catalyzed giving rise to a decrease in the Tafel slope from ca. 120 to less than 30 mV/dec. This is in contrast to the general consideration for noble metal catalysts such as Pt, where an increase in overpotential (or applied current density) results in an increase in Tafel slope. The latter finding is very interesting because it can potentially provide novel mechanistic insights into HER over non-noble metal catalysts.

I have a few rather minor comments mainly about the nature of the material that I append right below (also, most of my initial comments were answered by the existing thorough review comments).

Response. Thank you very much for your recommendations and comments. We appreciate the reviewer by commenting that our work “implement very good the concept of ‘structure engineering over catalytic activity’”, “is very interesting”, “provide novel mechanistic insights into HER”, etc.

Comment 1. In the preparation procedure, it is not clear how the authors adjusted the stoichiometry of the material since, the amount of Cu reacted is practically unknown (the reaction was carried out on bulk Cu with just MoS₂).

Response 1. Thank you very much for your comments. The reviewer is correct that the reaction happens between bulk Cu and few-layer MoS₂. In our experiments, we fixed the material preparation condition as below. We loaded 10 mg cm⁻² MoS₂ on a 0.4 mm thick copper foam with 95% porosity, followed by high temperature annealing at 750 °C for 2 h in an Ar (200 sccm) and H₂ (10 sccm) atmosphere. Under such reaction conditions, we found that pure Chevrel phase CuMo₆S₈ was obtained. We carefully compared the XRD patterns of as-prepared sample with other Chevrel phases (Cu_{1.84}Mo₆S₈, Cu₂Mo₆S₈, and Cu_{8.28}Mo₁₈S₂₄), and found that all the main characteristic peaks of our sample matched well with that of the CuMo₆S₈ but not others (Fig. R1). Therefore, we conclude that the as-prepared sample has the nearly same stoichiometry of CuMo₆S₈. Besides, we think it may be able to prepare Chevrel phases with different stoichiometry by controlling the loading of MoS₂, the thickness of Cu foam, annealing temperature and time.

Changes to revised SI. Fig. R1 has been added on Page 9 in the revised SI.

Figure R1. The comparison of XRD patterns between the as-prepared sample and different Chevrel phases, including CuMo_6S_8 , $\text{Cu}_{1.84}\text{Mo}_6\text{S}_8$, $\text{Cu}_2\text{Mo}_6\text{S}_8$, and $\text{Cu}_{8.28}\text{Mo}_{18}\text{S}_{24}$. This figure has been added on Page 9 in the revised SI.

Comment 2. In the TEM data it would be useful to provide the lattice spacing for the (101) and (111) planes and compare the results with what is reported in the literature for the CP (I personally haven't seen any data for the (131) plane). Furthermore, MoS_2 has an interplanar distance of ca. 0.27 nm. Can the authors prove that the interplanar distance ascribed to the (131) plane is not related to the presence of modified MoS_2 (e.g., Cu-doped)?

Response 2. Thank you very much for your instructive comments. To further distinguish CuMo_6S_8 and MoS_2 in interplanar distances, we have followed your suggestions and provided TEM images showing the interplanar distances of (101),

(110), (131), and (223) crystalline planes (Fig. R2). These four crystalline planes correspond well with the main characteristic peaks in XRD patterns (Fig. R3 below). In addition, we found a report about (131) plane in the literature (Electrochimica Acta, 2017, 258, 236-240), although it is indeed rarely reported as you pointed out. In our experiments, we did not find MoS₂ phase from XRD patterns (Figs. R3 and R4) and Raman spectra of the as-prepared sample (Fig. R5), which suggest that MoS₂ phase in the materials is negligible.

Changes to revised SI. Fig. R2 has been added on Page 8 in the revised SI.

Figure R2. TEM images of CuMo₆S₈ near the top surface (a) and near the interface between CuMo₆S₈ and Cu (b). The insets are corresponding FFT patterns. This figure was added on Page 8 in the revised SI.

Comment 3. In the XRD data there is a shift of the MoS₂ peak from 14.4 ° to 13.7 ° that the authors attribute to the conversion of MoS₂ to CP. In line with the above comment (#2), the unclear features in the XRD diffractogram (possibly related to the relatively thin thickness of the film) and considering the experimental procedure followed, how can the possibility of Cu-doped MoS₂ formation (instead of CP) be excluded?

Response 3. Thank you very much for your instructive comments. First, we re-performed the XRD characterization to obtain more clear data by collecting more amount of materials while removing the Cu foam substrate, and compared with the

MoS₂ sample before annealing (Fig. R3). There is a clear shift of 0.6° between (101) plane of CuMo₆S₈ (PDF #34-1379) and (002) plane of MoS₂ (PDF #37-1492). Besides, there are also other obvious differences between them, such as the absence of MoS₂ (100), (102) and (103) planes in the CuMo₆S₈ sample. Therefore, the possibility of the existence of MoS₂ can be excluded by XRD patterns.

Second, we compared the XRD patterns of as-prepared samples before annealing, and after annealing at different temperatures (250 °C, 550 °C, and 750 °C, as shown in Fig. R4). We found that when the temperature reaches 550 °C, the phase of CuMo₆S₈ begins to form, and the phase of MoS₂ basically disappears. When the temperature is further increased to 750 °C, the phase of MoS₂ completely disappears, and the crystallinity of CuMo₆S₈ increases, which further excludes the possibility of Cu-doped MoS₂.

Third, we carried out Raman characterization of the as-prepared CuMo₆S₈ sample (Fig. R5). We found that the typical Raman signals of Chevral phase appeared, such as E_g (126, 145, 360, 384 cm⁻¹) and A_g (202, 220, 285 cm⁻¹) peaks (Physical Review B, 1987, 36, 1952). Among them, the vibration peak at 405 cm⁻¹ corresponds to the typical Mo-S vibration (ACS Appl. Energy Mater. 2021, 4, 13015-13026). In contrast, based on literature (Nanoscale Adv., 2021, 3, 1747; Langmuir 2021, 37, 4847-4858; J. Phys. D: Appl. Phys. 2016, 49, 165003), Cu-doped MoS₂ only shows two vibration peaks of E_{12g}¹ and A_{1g} from MoS₂. These two MoS₂ related peaks will show certain shifts due to doping effect. This difference in Raman spectra of our CuMo₆S₈ sample and Cu-doped MoS₂ reported in literature clearly show that our sample is not modified MoS₂. Based on the above evidences, we can exclude the possibility of Cu-doped MoS₂.

Changes to revised manuscript and SI. Fig. R3 has been used as Fig. 1i on Page 6 in the revised manuscript. Fig. R4 and Fig. R5 have been added on Page 9 in the revised SI as Figs. S8 and S9.

Figure R3. X-ray diffraction patterns of the samples before and after annealing. This figure has been used as new Fig. 1i in the revised manuscript.

Figure R4. X-ray diffraction patterns of the samples before annealing, annealed at 250 °C, 550 °C, and 750 °C. This figure was added on Page 10 in the revised SI.

Figure R5. Raman spectra of the samples before and after annealing. This figure was added on Page 10 in the revised SI.

Comment 4. In the XPS spectrum of Mo 3d (5/2), there is a shift to lower binding energy values (ca. 2 eV) compared to what is reported in the literature for non-leached CP (for example in our paper). This might imply a charge transfer from the “doping/impurity” element (i.e., Cu) towards Mo. How this finding is compared to a Cu-doped MoS₂ sample?

Response 4. Thank you very much for your instructive comments. The Mo 3d_{5/2} of our sample locates at 227.9 eV, which is within the range of 226.8-228.3 eV of the reported Chevrel phase based on the NIST database (<https://srdata.nist.gov/xps/>) and from references (e.g., ACS Appl. Energy Mater., 2021, 4, 13015-13026; J. Catal., 1989, 117, 246; J. Catal., 93, 1985, 375). In contrast, the Mo 3d_{5/2} of MoS₂ mainly locates in the range of 228.8-230.2 eV, and there is a difference of about 2.0 eV between MoS₂ and Chevrel phase. Note that when Cu is doped into the MoS₂ lattice, the electron transfer effect may occur (Nanoscale Adv., 2021, 3, 1747; J. Phys. D: Appl. Phys., 2016, 49, 165003), causing a shift of Mo 3d_{5/2} to lower binding energy of 227.0-228.0 eV. Therefore, we adopted the above-mentioned XRD, Raman, and HRTEM techniques to exclude the existence of Cu-doped MoS₂ in our CuMo₆S₈ sample.

Comment 5. The method used for the determination of ECSA is not clear. I understand that the $40 \mu\text{F}/\text{cm}^2$ specific capacitance value for CP and MoS_2 is based on the theoretical capacitance for monolayer MoS_2 (which renders the approach highly approximate) but where does the value of $196 \mu\text{C cm}^{-2}$ for Pt come from?

Response 5. Thank you very much for your comments. As you pointed out, the specific capacitance for Chevrel phase and MoS_2 is based on the theoretical capacitance of monolayer MoS_2 ($30\text{-}60 \mu\text{F cm}^{-2}$). For MoS_2 and Chevrel phase, here the specific capacitance value of $40 \mu\text{F cm}^{-2}$ is used according to the theoretical capacitance in the references (e.g., J. Mater. Chem. A, 2016, 4, 6824; Mater. Today Chem., 2019, 14, 100207; Adv. Funct. Mater. 2018, 28, 1807086). For Pt, the specific capacitance value of $196 \mu\text{F cm}^{-2}$ is used according to the references (e.g., Electrochimica Acta, 2005, 50, 2469; Electrochimica Acta, 2017, 224, 468; Int. J. Electrochem. Sci., 2011, 6, 4454).

Changes to revised SI. The above references were added on Page 26 in the revised SI to make these points clear.

REVIEWER COMMENTS

Reviewer #1 (Remarks to the Author):

The manuscript is now much improved and complete for the publication. The authors elaborated to revised the manuscript supporting their strategy. The reviewer now suggest the manuscript can be published in Nature Comm. without any further revision.

Reviewer #5 (Remarks to the Author):

I think the authors have done a good job at responding to most of the points previously raised, which related to the characterization of the active material - namely showing that it was a Chevrel phase (and the particular phase identified) and not just a Cu-doped MoS₂ material. This is good.

There are still some problems, however. I do not understand the high capacitance value quoted (e.g from the Pickup paper, now added, Electrochimica Acta 50, 2469-2474 (2005)). The value quoted of 196 microF per cm² relates to a Pt/C electrode and must relate to the entire structure (Pt and C) and not just to the Pt. There are some geometrical artefacts, I think, in this value and the authors should compare with values for "Pure" (ie geometrically flat) Pt electrodes. Also the authors should give more detail as to the intrinsic reasons for the "better" catalysis they see (when they account for the eSCA in a "correct" way, as they claim) with their material, compared to Pt/C. The hydrogen adsorption energy for Pt is such that this material cannot be bettered for HER, so the factors relating to support interactions and bubble detachment must be the reasons for the improvement seen but these did not seem to be so clearly emphasized, at least on my reading of the work. Also, how do the authors measure a contact angle of zero degrees, as they report in the context of the bubble detachment work? There must be some imprecision associated with this data.

Response to Reviewer #1

Comment. The manuscript is now much improved and complete for the publication. The authors elaborated to revise the manuscript supporting their strategy. The reviewer now suggest the manuscript can be published in Nature Comm. without any further revision.

Response. Thank you very much for your recommendation.

Response to Reviewer #5

Comment. I think the authors have done a good job at responding to most of the points previously raised, which related to the characterization of the active material - namely showing that it was a Chevrel phase (and the particular phase identified) and not just a Cu-doped MoS₂ material. This is good.

Response. Thank you very much for your positive recommendations and helpful comments. We appreciate the reviewer for commenting that “the authors have done a good job at responding to most of the points previously raised”. We also thank the reviewer for pointing out that the active material characterization is now complete.

Comment 1. There are still some problems, however. I do not understand the high capacitance value quoted (e.g from the Pickup paper, now added, Electrochimica Acta 50, 2469-2474 (2005)). The value quoted of 196 micro F per cm² relates to a Pt/C electrode and must relate to the entire structure (Pt and C) and not just to the Pt. There are some geometrical artefacts, I think, in this value and the authors should compare with values for "Pure" (ie geometrically flat) Pt electrodes.

Response 1. Thank you very much for your comments. We agree with your opinion that the specific capacitance value quoted is contributed by both Pt and C. Actually, the reference sample we used is indeed 20 wt% Pt/C electrode, which is consistent with that reported in literature. That is why we use this quoted value of 196 $\mu\text{F cm}^{-2}$. Besides, we also take your suggestions and use the classical specific capacitance value of pure

Pt ($40 \mu\text{F cm}^{-2}$, J. Electrochem. Soc., 1992, doi.org/10.1149/1.2069469, J Mater. Chem. A, 2016, doi.org/10.1039/C6TA07009D) to normalize the pure Pt foil electrode ($1 \times 1 \text{ cm}^2$), as shown in Fig. R1b and 1c below. The double layer capacitance of the pure Pt flat foil is much smaller than nanostructured electrodes (e.g., Pt/C), thus showing better performance than Pt/C electrode at ECSA normalized LSV curves, which is consistent with the previous literature (Fundamental research, 2022, doi.org/10.1016/j.fmre.2022.03.017). It is well accepted that Pt has the best intrinsic HER activity, most reported catalysts (including our material) cannot surpass it in terms of intrinsic activity. For large current density HER performance, it is related to not only intrinsic activity but also other factors including mass transfer and morphology optimization, that is the reason why flat Pt electrode shows poorer performance than our material (Fig. R1a.). That is the concept expressed in our work that “structure engineering is important in improving catalytic activity at large current density”.

Changes to revised SI. The experimental details about flat Pt foil and Fig. R1 have been added on Page 2 and Page 11 in the revised SI.

Figure R1. LSV curves (a), double layer capacitance per unit area (b) and ECSA normalized LSV curves of the CuMo₆S₈, MoS₂, Pt/C and flat Pt foil electrodes. Figure R1 has been added as Supplementary Fig. 11 on Page 11 in the revised SI.

Comment 2. Also the authors should give more detail as to the intrinsic reasons for the "better" catalysis they see (when they account for the ESCA in a "correct" way, as they claim) with their material, compared to Pt/C. The hydrogen adsorption energy for Pt is such that this material cannot be bettered for HER, so the factors relating to support interactions and bubble detachment must be the reasons for the improvement seen but these did not seem to be so clearly emphasized, at least on my reading of the work.

Response 2. Thank you very much for your instructive comments which we totally agree. We are glad to accept your suggestions and have emphasized the key factors (including support interaction, bubble detachment, etc) affecting the HER performance on Pages 7, 13 and 14 of revised manuscript, as follows. “For hydrogen evolution at large current density, interfacial charge transfer between catalyst and support significantly affects catalytic kinetics. Electrochemical impedance spectra (EIS) show that the charge transfer resistance of CuMo₆S₈/Cu is twice and three times less than that of the MoS₂ and Pt/C electrodes. (Supplementary Fig. 11a). This is because the interfacial Schottky barrier has been eliminated in CuMo₆S₈/Cu due to the transform from semiconductor (MoS₂) to metal (CuMo₆S₈) (Fig. 5f). Besides, charge transfer resistance can be reduced further because we do not use Nafion binder compared with Pt/C electrode.” “Moreover, HER performance at large current density is largely affected by gas bubble detachment. The growth and accumulation of gas bubbles on the catalyst surface would produce micro-convection, impede ion transport and block active site, causing extra energy consumption namely transport overpotential (η_{trans})^{44, 45}. The results of η_{trans} at different current densities are plotted in Supplementary Fig. 23 and detailed derivations are shown in Supplementary Note 2. Obviously, the η_{trans} and its increasing trend of the CuMo₆S₈/Cu electrode at large current density is much smaller than that of Pt/C. This is because the CuMo₆S₈/Cu electrode has a small catalyst-bubble interfacial adhesion force, thus exhibits faster bubble evolution kinetics.”

Changes to revised manuscript. The modified part has been added on Pages 7, 13 and 14 of revised manuscript.

Comment 3. Also, how do the authors measure a contact angle of zero degrees, as they report in the context of the bubble detachment work? There must be some imprecision associated with this data.

Response 3. The contact angle test was performed on KRUSS DSA30 which was monitored by an optical microscope, and the detailed process was shown in Fig. R2. When the liquid droplet touched the sample, it spread quickly on the surface of sample and almost wetted the CuMo₆S₈ electrode, showing the superhydrophilic state.

However, the limited precision of optical microscopy cannot give specific values of contact angle in the case of near zero degrees, so to be more precise, the contact angle is “near zero degrees”. Besides, we attributed the superhydrophilic state to high textured surface and metal intrinsic of the CuMo_6S_8 electrode. It is the superhydrophilic surface that makes the bubbles extremely easy to detach from the electrode and results in a $\sim 0^\circ$ contact angle, which improves performance of the CuMo_6S_8 electrode at large current density.

Changes to revised SI. The expression of “0 °” and “zero degrees” have been changed to “ $\sim 0^\circ$ ” and “near zero degrees” in the revised manuscript.

Figure R2. The snapshot of contact angle test of the $\text{CuMo}_6\text{S}_8/\text{Cu}$ electrode. The arrows represent move direction of the sample. The sample gradually moves toward the droplet (Fig. R2 a) and touches it (Fig. R2b). Afterwards, the sample is gradually wetted by the droplet (Fig. R2c) until it is completely wetted (Fig. R2d), showing the superhydrophilic state and giving a contact angle of $\sim 0^\circ$. The interval between each photo is about 2s.

REVIEWER COMMENTS

Reviewer #5 (Remarks to the Author):

The authors have clarified some things but the quoted capacitance values are still poorly justified, in my opinion.

What is the electrolyte? The sentence added to the SI:

" The specific capacitances of CuMo₆S₈ and MoS₂ are 0.04 mF cm⁻² 1-4

, and

that of Pt/C and pure Pt are 0.196 mF cm⁻² 5-6

and 0.04 mF cm⁻² 7-8"

needs to refer to a common electrolyte phase.

I also do not think the values quote are correct. If one follows ref 2 of the SI (Lai et al) for example, they cite the following work: <https://doi.org/10.1021/acsami.5b03399> (Benson et al) where they quote the specific capacitance of pristine MoS₂ to be 0.06 mF cm⁻². Basically, the authors should perform a systematic study of the capacitances of their materials, making proper comparison with the appropriate bulk materials. I do not believe the values quoted at present, I think they are not meaningful.

Response to Reviewer #5

Comment. The authors have clarified some things but the quoted capacitance values are still poorly justified, in my opinion. What is the electrolyte? The sentence added to the SI: “The specific capacitances of CuMo_6S_8 and MoS_2 are 0.04 mF cm^{-2} ¹⁻⁴, and that of Pt/C and pure Pt are 0.196 mF cm^{-2} ⁵⁻⁶ and 0.04 mF cm^{-2} ⁷⁻⁸” needs to refer to a common electrolyte phase. I also do not think the values quote are correct. If one follows ref 2 of the SI (Lai et al) for example, they cite the following work: <https://doi.org/10.1021/acsami.5b03399> (Benson et al) where they quote the specific capacitance of pristine MoS_2 to be 0.06 mF cm^{-2} . Basically, the authors should perform a systematic study of the capacitances of their materials, making proper comparison with the appropriate bulk materials. I do not believe the values quoted at present, I think they are not meaningful.

Response. Thank you very much for your comments. Our electrolyte is 1M KOH. We agree that the kind of electrolyte may affect the specific capacitance of materials. To systematically study the specific capacitances of MoS_2 -based and Pt-based materials in different electrolytes, we conducted an exhaustive literature search, as shown in Table R1. **For MoS_2 -based materials**, about 60% and 30% of the literature use 40 and 60 $\mu\text{F cm}^{-2}$ as the specific capacitance values, respectively. Amongst, the value of 40 $\mu\text{F cm}^{-2}$ is widely used in alkaline electrolytes. We noticed that quite a few literature explaining that they choose this average value of 40 $\mu\text{F cm}^{-2}$ because “the specific capacitance for a flat surface was generally found to be in the range of 20-60 $\mu\text{F cm}^{-2}$.” (e.g., Joule, 2019, 3, 2955-2967; Adv. Mater., 2018, 30, 1707105; Chem. Eng. J., 2021, 409, 128158). Therefore, it is reasonable for us to use 40 $\mu\text{F cm}^{-2}$ for MoS_2 -based materials (MoS_2 and CuMo_6S_8) in 1M KOH electrolyte.

For Pt-based materials, the reported specific capacitances are mostly between 17 and 60 $\mu\text{F cm}^{-2}$ both in acid and alkaline electrolytes. The value of 196 $\mu\text{F cm}^{-2}$ cited before is larger probably because the electrode used in the literature (Pt/C containing $\text{Os}(\text{bpy})_3^{2+}$, Electrochimica Acta, 2005, 50, 2469-2474) is different from ours. As

mentioned by Jaramillo et al., “specific capacitances have been measured for a variety of metal electrodes (e.g., Pt, Ni, Co, Cu, Mo) in acidic and alkaline solutions and typical values reported range between $C_s = 0.015\text{-}0.110\text{ mF cm}^{-2}$ in H_2SO_4 and $C_s = 0.022\text{-}0.130\text{ mF cm}^{-2}$ in NaOH and KOH solutions.” (J. Am. Chem. Soc. 2013, 135, 16977-16987). Based on typical reported values, they suggested that “using general specific capacitances of $C_s = 35\text{ }\mu\text{F cm}^{-2}$ in 1 M H_2SO_4 and $C_s = 40\text{ }\mu\text{F cm}^{-2}$ in 1 M NaOH.” Besides, the average value (c.a. $39.1\text{ }\mu\text{F cm}^{-2}$) of reported literature is close to $40\text{ }\mu\text{F cm}^{-2}$, so we think using the value of $40\text{ }\mu\text{F cm}^{-2}$ for Pt-based materials (Pt/C and Pt foil) in 1M KOH electrolyte is acceptable.

In general, the specific capacitance value used in most literature is an average value within a reasonable range. $40\text{ }\mu\text{F cm}^{-2}$ is a “typical” value for most metal-based materials like Pt in alkaline electrolyte. We think using such a generally accepted value will facilitate the comparison among literature.

Changes to revised SI. The ECSA normalized LSV curves, and related references of specific capacitance have been updated on Pages 3, 11 and 26 in the revised SI.

Table R1. The specific capacitances reported in MoS_2 -based and Pt-based materials.

Materials	Electrolytes	Specific capacitance ($\mu\text{F cm}^{-2}$)	References
MoS ₂ -based	0.5M H ₂ SO ₄	60	Mater. Today Chem., 2019, 14, 100207
	0.5M H ₂ SO ₄	60	ACS Appl. Mater. Interfaces 2015, 7, 25, 14113–14122
	0.5M H ₂ SO ₄	60	Nat. Mater., 2012, 11, 963–969
	0.5M H ₂ SO ₄	60	J. Mater. Chem. A, 2016, 4, 6824-6823
	0.5M H ₂ SO ₄	60	ACS Appl. Mater. Interfaces, 2020, 12, 35995–36003
	0.5M H ₂ SO ₄	35	Adv. Funct. Mater., 2018, 28, 1807086
	0.5M H ₂ SO ₄	40	Nanoscale Res. Lett., 2021, 16, 137
	0.5M H ₂ SO ₄	35	Appl. Catal. B: Environ., 2019, 258, 117964
0.5M H ₂ SO ₄	40	Appl. Catal. B: Environ., 2021, 284, 119708	

	0.5M H ₂ SO ₄	60	Carbon, 2020, 158, 216-225
	0.5M H ₂ SO ₄	60	ACS Appl. Mater. Interfaces, 2016, 8, 8, 5517–5525
	0.5M H ₂ SO ₄	40	Chem. Eng. J., 2022, 428, 132072
	0.5M H ₂ SO ₄	40	J. Environ. Chem. Eng., 2022, 10,108038
	0.5M H ₂ SO ₄	35	ACS Sustainable Chem. Eng. 2020, 8, 11, 4547–4554
	0.5M H ₂ SO ₄	40	Nat. Commun., 2021, 12, 709
	0.5M H ₂ SO ₄	40	ACS Appl. Mater. Interfaces 2022, 14, 16338–16347
	1M KOH	20	Int. J. Hydrogen Energy, 2020, 45, 9773–9782
	1M KOH	40	ACS Sustainable Chem. Eng., 2020, 8, 11, 4547–4554
	0.5 M KOH	40	Joule, 2019, 3, 2955-2967
	1M KOH	40	J. Mater. Sci., 2020, 55, 16197–16210
	1M KOH	40	Adv. Mater., 2018, 30, 1707105
	1M KOH	40	Adv. Energy Mater., 2018, 8, 1801345
	1M KOH	40	Chem. Eng. J., 2022, 428, 131055
	1M KOH	40	ACS Appl. Mater. Interfaces, 2020, 12, 40194–40203
	pH universal	40	Adv. Funct. Mater., 2020, 30, 2002536
	pH universal	40	ACS Catal., 2021, 11, 4486–4497
	pH universal	40	Chem. Eng. J., 2021, 409, 128158
	pH universal	40	Nat. Commun., 2021,12, 6776
	Pt: 0.5M H ₂ SO ₄	60	ACS Appl. Nano Mater., 2022, 5, 1377–1384
	Pt: 1 N KOH	60	Electrochim. Acta, 1972, 17, 2249-2265
Pt-based	Pt: 1 M KOH	28	J. Chem. Soc., Faraday Trans., 1993, 89, 235-242
	Pt/C: 1M KOH	30	J. Electrochem. Soc., 1996, 143, 919-926
	Pt: 1 M H ₂ SO ₄	17	J. Electrochem. Soc., 1969, 116, 1112-1116.
	Pt: 1 N H ₂ SO ₄	35-45	J. Electrochem. Soc., 1979, 126, 424-430